# *LV*-Eval: A Balanced Long-Context Benchmark with 5 Length Levels Up to 256K

**Tao Yuan[2], Xuefei Ning[1,†], Dong Zhou[2], Zhijie Yang[2], Shiyao Li[1], Minghui Zhuang[2], Zheyue Tan[2], Zhuyu Yao[2], Dahua Lin[3,4], Boxun Li[2,*], Guohao Dai[2,5,†], Shengen Yan[1,2], Yu Wang[1,†]**

[1] Tsinghua University, [2] Infinigence-AI, [3] Shanghai Artificial Intelligence Laboratory
[4] The Chinese University of Hong Kong, [5] Shanghai Jiao Tong University

## Abstract

State-of-the-art large language models (LLMs) are now claiming remarkable supported context lengths of 256*k* or even more. In contrast, the average context lengths of mainstream benchmarks are insufficient (5*k*-21*k*), and they suffer from potential knowledge leakage and inaccurate metrics, resulting in biased evaluation. This paper introduces *LV*-Eval, a challenging long-context benchmark with five length levels (16*k*, 32*k*, 64*k*, 128*k*, and 256*k*) reaching up to 256*k* words. *LV*-Eval features two main tasks, single-hop QA and multi-hop QA, comprising 11 bilingual datasets. The design of *LV*-Eval has incorporated three key techniques, namely confusing facts insertion (CFI), keyword and phrase replacement (KPR), and keyword-recall-based metric design. The advantages of *LV*-Eval include controllable evaluation across context lengths, challenging test instances with confusing facts, mitigated knowledge leakage, and more objective evaluation. We evaluate 15 LLMs on *LV*-Eval and conduct ablation studies on the benchmarking techniques. The results reveal that: (i) Moonshot-v1 and recent large-scale open-source models, such as Qwen-2.5-72B and Llama-3.1-70B, achieve the highest performance on *LV*-Eval, particularly at lengths below 64*k*. (ii) Models exhibit distinct score trends. For example, GLM-4-9B-128k, Yi-6B-200k, and Llama3-8B-1M exhibit a relatively gentle degradation of performance, but their absolute performances may not necessarily be higher than those of LLMs with shorter context lengths. (iii) LLMs' performances can significantly degrade in the presence of confusing information, especially in the pressure test of "needle in a haystack". (iv) Issues related to knowledge leakage and inaccurate metrics introduce bias in evaluation, and these concerns are alleviated in *LV*-Eval.

## 1 Introduction

Large language models (LLMs) have demonstrated exceptional performance on a variety of natural language processing tasks. The ability of long-context understanding is crucial for LLMs to deal with tasks based on longer contexts, such as books, lengthy chat history, and so on. Recently, extensive efforts have been devoted in enlarging the supported context length (i.e., the number of tokens that the model can accept as input) of LLMs. These efforts have pushed the supported context length of LLMs from 2*k* tokens to 32*k* tokens (Touvron et al., 2023; Zheng et al., 2023; Bai et al., 2023a; Zeng et al., 2022; Li et al., 2023a), and some models have achieved a remarkable context length of 128*k* and 200*k* (Peng et al., 2023a; Achiam et al., 2023; Yi, 2023).

In contrast to the rapid evolution of the models' supported context length, existing benchmarks have lagged behind. The average word count in current long-context benchmarks

---

*Project leader

†Correspondence to Yu Wang <yu-wang@tsinghua.edu.cn>, Xuefei Ning <foxdoraame@gmail.com>, Guohao Dai <daiguohao@sjtu.edu.cn, daiguohao@infini-ai.com>.

| Benchmark | #Datasets | Avg #Words | Min/Max Words | Length Levels | Opt. Metric | Lang. |
|---|---|---|---|---|---|---|
| ZeroSCROLLS Shaham et al. (2023) | 10 | 13,556 | 1,023/320,774 | none | | en |
| LooGLE Li et al. (2023b) | 7 | 21,247 | 10,927/246,182 | none | | en |
| L-Eval An et al. (2023) | 20 | 12,993 | 2,119/170,256 | none | ✓ | en |
| BAMBOO Dong et al. (2023) | 10 | 5,067 | 229/14,858 | 4k,16k | | en+zh |
| LongBench Bai et al. (2023b) | 21 | 9,486 | 128/71,954 | 0-4k,4k-8k,8k+ | | en+zh |
| *LV*-Eval | 11 | 102,380 | 11,896/387,406 | 16k,32k,64k,128k,256k | ✓ | en+zh |

Table 1: Comparison of different long-context benchmarks. We count the number of words for the English datasets and the number of characters for the Chinese datasets. The punctuation marks are taken into account, while tabs, blank spaces, and newlines are not included.

typically falls within the range of 32*k* (Bai et al., 2023b; Li et al., 2023b; An et al., 2023; Shaham et al., 2023; Dong et al., 2023), considerably shorter compared to the supported context lengths of state-of-the-art long-context models. Moreover, previous benchmarks primarily consist of *unaltered* public documents and articles. This could be problematic for two reasons: (i) the data might be involved in LLMs' training processes, and (ii) the facts within them might be common-sense facts found in other training resources. The presence of this issue, known as "knowledge leakage" (Zhou et al., 2023a), can lead to models answering questions with memorization or common-sense knowledge instead of understanding long-range contexts. Last but not least, the automatic metrics employed in most of the existing benchmarks are susceptible to the variations in answer format and the inclusion of irrelevant words. Such metrics struggle to accurately assess the answer quality.

To address these issues, we propose *LV*-Eval, a bilingual benchmark with up to 256*k* words. *LV*-Eval incorporates distractions and confusions to make the test more challenging, replaces keywords and rephrases sentences to prevent knowledge leakage, and employs a more accurate metric. We summarizes the key characteristics of *LV*-Eval as follows:

- **Sufficiently long context length to evaluate state-of-the-art models**: *LV*-Eval comprises 5 length levels with word counts of 16*k*, 32*k*, 64*k*, 128*k*, and 256*k*. Test instances across these levels share the same set of question-answer (QA) pairs, and only differ in the context content and length. Testing on the same QA pairs with different context lengths facilitates a controllable evaluation of models' long-context ability.

- **Incorporation of distraction and confusion to increase difficulty**: When constructing the context for each test instance, we mix up distracting documents and supporting documents. This approach evaluates the model's ability in pinpointing key information in a large bunch of distracting texts. In addition, we insert confusing facts generated by GPT-4 and revised by human annotators into the context. This assesses the model's capability to accurately reason in the presence of interference.

- **Keyword and phrase replacement to mitigate knowledge leakage**: To mitigate the biased evaluation of long-context ability caused by knowledge leakage, we replace the keywords and phrases in the context and QA pairs. The replacement rules are annotated by human annotators. In this way, *LV*-Eval requires LLMs to rely on the understanding of context to answer questions rather than relying on memorization or common-sense knowledge.

- **Keyword-recall-based metric for more objective scoring**: Existing *N*-gram metrics such as the F1 score are sensitive to the format variations and non-informative words in the answer, which results in inaccurate scores. To address this, we manually annotate answer keywords and a blacklist of unrelated words. The golden answers are the critical words or sentences extracted from original ground-truth (GT) answers, while the word blacklist contains common and non-informative words such as 'the', 'a', 'of', and so on. The metric calculation follows a two-stage procedure: the first stage calculates the recall of golden answer keywords. if the recall exceeds a certain threshold, the second stage will remove all the blacklisted words and then calculate the F1 score between the prediction and the GT answer. This metric design can get scores with higher objectivity.

| Task | Dataset | CFI | #KPR | AK | Language | #QA pairs | #Contexts |
|---|---|---|---|---|---|---|---|
| Single-hop QA | **lic-mixup** | ✓ | | ✓ | zh | 197 | 985 |
| | **loogle-SD-mixup** | | | ✓ | en | 160 | 800 |
| | **cmrc-mixup** | | 786 | | zh | 200 | 1,000 |
| | **multifieldqa-en-mixup** | ✓ | 476 | ✓ | en | 101 | 505 |
| | **multifieldqa-zh-mixup** | ✓ | 424 | ✓ | zh | 133 | 665 |
| | **factrecall-en** | ✓ | 3 | ✓ | en | 1 | 200×5 |
| | **factrecall-zh** | ✓ | 3 | ✓ | zh | 1 | 200×5 |
| Multi-hop QA | **dureader-mixup** | | | | zh | 176 | 880 |
| | **loogle-CR-mixup** | | | ✓ | en | 99 | 495 |
| | **loogle-MR-mixup** | | | ✓ | en | 139 | 695 |
| | **hotpotwikiqa-mixup** | ✓ | 232 | ✓ | en | 124 | 620 |

Table 2: Data statistics of *LV*-Eval. The abbreviations "CFI", "KPR", "AK" stand for "Confusing Fact Insertion", "Keyword and Phrase Replacement", and "Answer Keywords", respectively. "#KPR" is the number of KPR rules. Note that in **factrecall-en** and **factrecall-zh**, all QA pairs are the same across all test instances, i.e., there is only one unique QA pair for each of the two datasets.

**Findings.** We evaluate 15 LLMs on *LV*-Eval and summarize the main findings as follows: (i) Moonshot-v1 and recent large-scale open-source models, such as Qwen-2.5-72B and Llama-3.1-70B, achieve the highest performance on *LV*-Eval, particularly at lengths below 64*k*. (ii) Models exhibit distinct score trends. For example, GLM-4-9B-128k, Yi-6B-200k, and Llama3-8B-1M exhibit a relatively gentle degradation of performance, but their absolute performances may not necessarily be higher than those of LLMs with shorter context lengths. (iii) LLMs' performances can significantly degrade in the presence of confusing information, especially in the pressure test of "needle in a haystack". (iv) Issues related to knowledge leakage and inaccurate metrics introduce bias in evaluation, and these concerns are alleviated in *LV*-Eval.

## 2 *LV*-Eval Benchmark

*LV*-Eval focuses on two types of QA tasks: single-hop QA and multi-hop QA, and is comprised of 11 QA datasets (6 in English and 5 in Chinese). The data statistics for *LV*-Eval are outlined in Table 2. Each test instance in *LV*-Eval comprises three parts: a context (*C*), a question (*Q*), and a GT answer (*A*), where *C* is a synthetic document containing the information required to answer *Q*.

Datasets in *LV*-Eval are constructed with existing public datasets as the source, except for factrecall-en and factrecall-zh, which are constructed using the data from *PG19* (Rae et al., 2019) dataset and *Journey to the West* book. Each dataset consists of five subsets of different lengths: 16*k*, 32*k*, 64*k*, 128*k*, and 256*k*. All five subsets share the same question-answer (QA) pairs, meaning there are five contexts of varying lengths for each QA pair. This allows for a controllable evaluation of models' long-context ability when testing the same set of questions with different context lengths. In total, *LV*-Eval comprises 1,729 QA pairs and 1,729×5 = 8,645 synthetic contexts.

Figure 1 illustrates the construction process of *LV*-Eval. For **factrecall-en** and **factrecall-zh**, we write one QA pair for each dataset. For the rest 9 out of the 11 datasets, we first choose a specific number of QA pairs from existing QA datasets (Section 2.1). Then, for each unique QA pair, we go through three procedures to construct the context (Section 2.2):

1. **Context mixing up** (Section 2.2.1): We first construct five contexts of different lengths by mixing up supporting documents corresponding to the QA pair and several distracting documents. For **factrecall-en** and **factrecall-zh**, we mix the supporting evidence of the single QA pair with distracting documents from two books. For other datasets, the distracting documents are unrelated to the question and are chosen from the context documents corresponding to non-selected QA pairs in the same source dataset.

2. **Confusing Facts Insertion (CFI)** (Section 2.2.2): Then, in some datasets, we introduce confusing facts by generating them with GPT-4, manually revising them, and randomly inserting these into the context. These confusing facts bear similarities to the original supporting facts but are factually different, without contradicting the original information. This helps make the test instances more challenging.

3. **Keyword and Phrase Replacement (KPR)** (Section 2.2.3): Finally, to reduce the impacts of knowledge leakage on evaluation results, we manually replace some keywords and phrases in the context and the QA pairs.

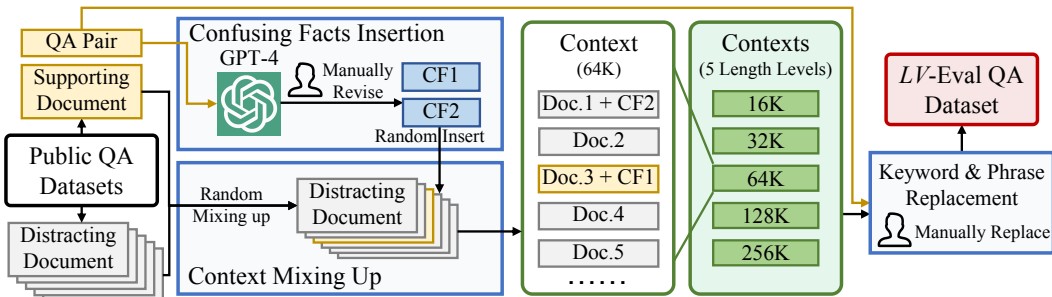

Figure 1: The construction process of *LV*-Eval. "CF" is short for "Confusing Fact".

When evaluating the generated answer, to mitigate the bias in existing metrics, we manually annotate the keywords in the GT answer and adjust the metric to focus more on the keywords (Section 2.3).

During the benchmark construction, we employ and guide five human annotators to revise the confusing facts, replace the keywords and phrases, and annotate the keywords in GT answers. These human annotators include three master students involved in LLM research and two master students majored in linguistics. The details of our annotation process are introduced in Appendix D.

## 2.1 Data Source and QA Pair Construction

We construct 11 datasets (see Table 2) using public data sources, including Long-instruction-en2zh (yuyijiong, 2023), HotpotQA (Yang et al., 2018), 2WikiMultihopQA (Ho et al., 2020), DuReader (Tang et al., 2020), LooGLE (Li et al., 2023b), LongBench (Bai et al., 2023b), CMRC 2018 (Cui et al., 2018), MultiFieldQA (Bai et al., 2023b), PG-19 (Rae et al., 2019) and the book of *Journey to the West*. The construction of QA pairs in each dataset is elaborated in Appendix C.

## 2.2 Context Construction

### 2.2.1 Context Mixing Up

Can the LLMs identify the key evidences to answer the target question within a long context? To assess this ability, as shown in Figure 1, *LV*-Eval randomly mixes the supporting documents with various distracting documents to generate five contexts of varying length for a given QA pair. For 9 out of the 11 datasets (excluding **factrecall-en** and **factrecall-zh**), the distracting documents are chosen from the contexts corresponding to the non-selected QA pairs in the source dataset. For **factrecall-en** and **factrecall-zh**, the distracting documents are extracted from the *PG-19* dataset and the book of *Journey to the West*.

For each length level, we sample distracting documents one by one until the cumulative word count meets the desired length level. Then, we shuffle the supporting and distracting documents, prepend a string "Passage i\n" to the *i*-th document, and concatenate them to form the final context.

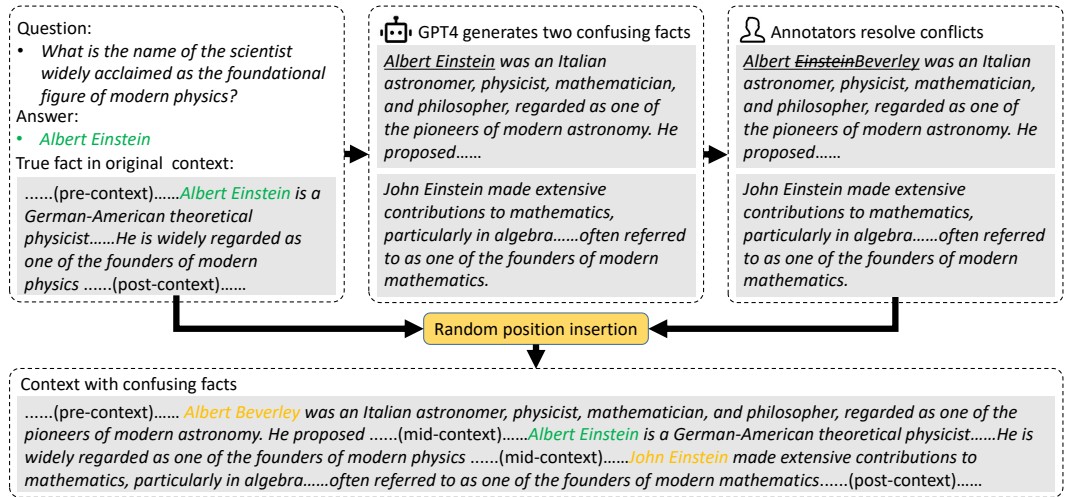

Figure 2: Steps for CFI. Firstly, we prompt GPT-4 to generate two descriptions that are close to the original fact. Then we ask human annotators to resolve any conflicts in the generated facts. For example, the first generated confusing fact "Albert Einstein was an Italian astronomer" is in conflict with the original fact and the human annotator revise it to "Albert Beverley was an Italian astronomer". Finally, the confusing facts are inserted into a randomly position in the context.

Note that in **hotpotwikiqa-mixup** and **dureader-mixup**, where multiple supporting documents exist for each QA pair, instead of regarding the multiple supporting documents a single unit, we disperse and shuffle all supporting and distracting documents.

### 2.2.2   Confusing Facts Insertion

Can the LLMs identify the key evidences correctly if there are confusing facts in the context? To assess this ability, we apply CFI in **hotpotwikiqa-mixup**, **lic-mixup**, **multifieldqa-en-mixup**, **multifieldqa-zh-mixup**, **factrecall-en**, and **factrecall-zh**, which inserts similar, factually different, non-contradictory facts into the context. These facts might mislead less meticulous models, leading them to generate incorrect answers.

The generation process of the confusing facts goes as follows. Firstly, we use the question and answer as the input, and prompt GPT-4 (Achiam et al., 2023) to generate two descriptions that are close to the original fact. The prompt for GPT-4 is shown in Figure A7. Then, we ask human annotators to resolve any conflicts in the generated facts. As illustrated in Figure 2, the generated confusing fact "Albert Einstein was an Italian astronomer" is in conflict with the original fact. Therefore, the human annotator revise it to "Albert Beverley was an Italian astronomer". After this generation and revising process, we insert the confusing facts into a randomly picked position between two sentences in the context.

### 2.2.3   Keyword and Phrase Replacement

Knowledge leakage is an important concern in LLM evaluation (Zhou et al., 2023a). On the one hand, the test data are usually collected from open-access sources, and we cannot fully rule out the possibility of their being involved in some LLMs' training process. On the other hand, some common-sense questions can be answered without referencing the provided context. Consequently, LLMs might rely on memorization and common-sense knowledge to answer the questions rather than fully understanding the context. This will cause inflated benchmark scores to overrate the long-context ability of models.

To mitigate the influences of knowledge leakage on the evaluation results, we conduct KPR according to manually crafted rules in **hotpotwikiqa-mixup**, **cmrc-mixup**, **multifieldqa-en-mixup**, **multifieldqa-zh-mixup**, **factrecall-en**, and **factrecall-zh**. Specifically, given a

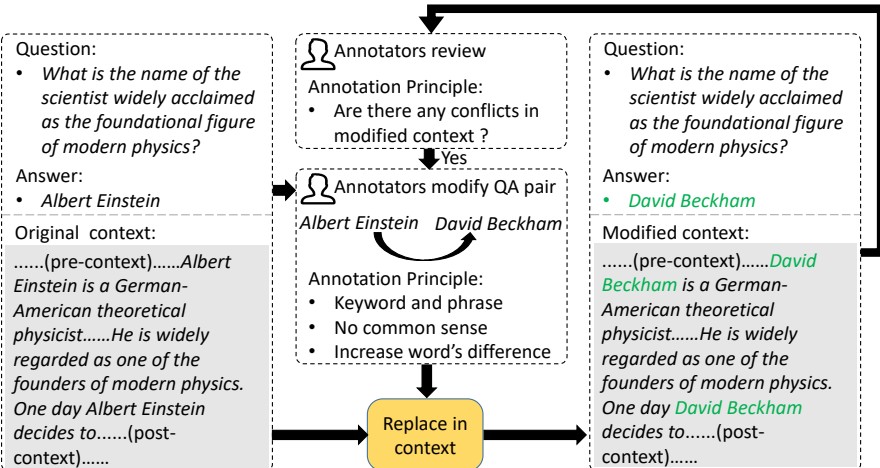

Figure 3: Steps for KPR. First, given a QA pair, the annotators are asked to select keywords or phrases to replace and write a substitute for each. Then, the selected keywords and phrases are replaced throughout the context and QA pair. Finally, annotators will check the modified context. If there is any conflict, the annotators are asked to revise the replacement rule until all conflicts are resolved.

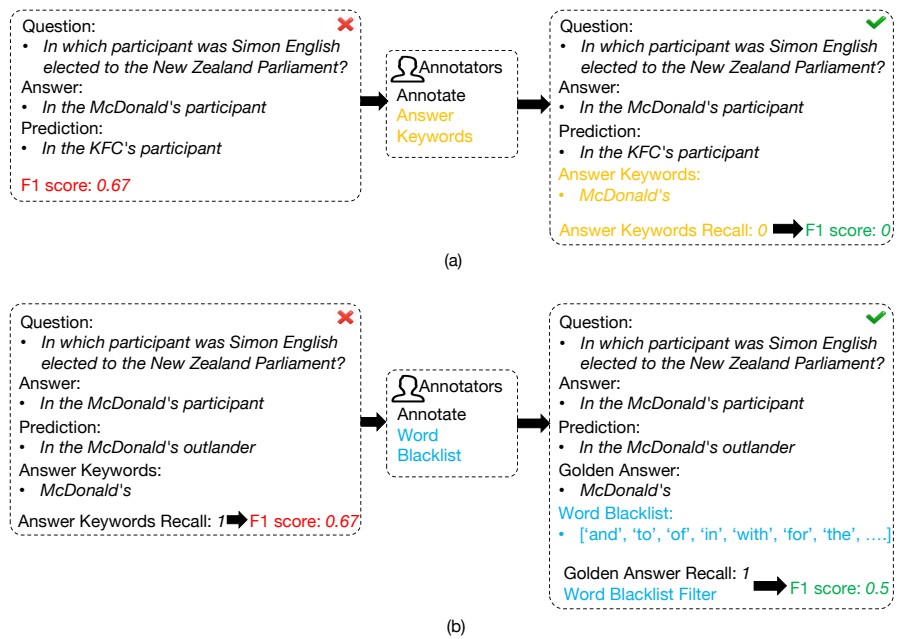

Figure 4: Keyword-recall-based two-stage metric calculation. (a) The vanilla F1 score (red) is inflated high. With the keyword recall-based metric, the final score is set to zero due to the low recall of answer keywords. (b) The vanilla F1 score (red) is inflated high due to irrelevant words. By filtering of blacklisted words, the final score is better calibrated.

QA pair, the annotators are asked to select keywords or phrases for replacement and write a substitute for each. After the selected keywords and phrases are replaced throughout the entire context, the annotators review the modified context to check and resolve any conflicts: If there are conflicts, the annotators are asked to revise the replacement rule until all conflicts are resolved. One example of the KPR process is shown in Figure 3. See Table 2 for the statistics of the number of replacement rules.

### 2.3 Metric Design

The quality evaluation of natural language generation is challenging. Current $N$-gram metrics, such as the F1 score, treat all words equally. The neglect of differences in word importance leads to evaluation bias. For example, in the sentence "Attention is all you need", the word "attention" carries the key information and is more important. However, the answer "Attention matters" will get a lower score than the answer "CNN is all you need", which is not what we expected. To this end, we adopt a two-stage metric calculation process.

Specifically, to evaluate an answer $A'$, we first calculate the recall of several "answer keywords" in $A'$. When the recall exceeds a certain threshold (0.2 for Chinese dataset, 0.4 for English datasets), we calculate the F1 score between $A'$ and GT answer $A$ as the final score for $A'$. otherwise, $A'$ gets a zero score. We manually annotate the answer keywords in the GT answer $A$ for **hotpotwikiqa-mixup**, **lic-mixup**, **loogle-CR-mixup**, **loogle-MR-mixup**, **loogle-SD-mixup**, **multifieldqa-en-mixup**, and **multifieldqa-zh-mixup**. Figure 4 (a) shows an example, demonstrating how this two-stage calculation helps avoid some inflated high evaluation scores.

When calculating the F1 score between $A'$ and $A$ in the second stage, we exclude common but non-informative words like 'the,' 'a', 'of', and so on. The word blacklist is constructed as follows. We first summarized the word counts in the generations of Llama2-7B-Chat-hf and ChatGLM3-6B-32K on all datasets and chose the top 100 words that matched the GT answer most frequently. Then, we manually annotate the non-informative words from the 100 words to construct the blacklist. Figure 4 (b) shows an example of how the word blacklist aids in calibrating the evaluation scores.

## 3 Evaluation

**Models and Inference.** We evaluate 3 commercial and 12 open-source LLMs on *LV*-Eval. Their information is summarized in Table A4. We follow the official implementation of all LLMs to conduct their inferences. Greedy sampling is used for generating tokens. For LLMs with a context window size smaller than the length of the data context, we truncate the data context in the middle, and concatenate the head and the tail of the context as input, ensuring that the QA instructions are fully contained within the input.

**Metrics.** For all tasks except **dureader-mixup** and **cmrc-mixup**, we evaluate the generated answers with our keyword-recall-based F1 metric, utilizing the annotated answer keywords and word blacklist. For **cmrc-mixup**, we omit the manual annotation of answer keywords since the answers in this dataset is already concise. Therefore, we use the F1 metric with word blacklist. In the case of **dureader-mixup**, where the GT answer lengths are relatively long, we do not manually annotate the answer keywords and use the ROUGH-L metric with the word blacklist.

### 3.1 Compare LLMs on *LV*-Eval

Figure 5 (a) shows the average scores across all 11 datasets of 15 LLMs at different length levels. We can see that (i) The commercial Moonshot-v1 and recent large-scale open-source models (Qwen2.5-72B, Llama-3.1-70B) achieve the best performances. Notably, both Qwen2.5-72B and Moonshot-v1 obtain average scores exceeding 40 at $16k$ and $32k$ lengths. (ii) Among open-source models with parameter sizes in the 6-9B range, GLM-4-9B achieves the best performance, even outperforming Llama-3.1-70B on longer lengths ($128k$, $256k$). (iii) Models exhibit distinct score trends, resulting in different relative rankings across different length levels. For example, the model with the largest context window size, Llama3-8B-1M, exhibits one of the slowest decline of performance from $16k$ to $128k$. Specifically, its scores at the length level $16k$ is lower than ChatGLM3-6B-32k and BlueLM-7B-32k-Chat. Nevertheless, as the length of input context increases, Llama3-8B-1M retains a higher score than these two models that need to truncate the input context. The similar phenomenon can be observed between Yi-6B-200k and two GPTs, as well as between GLM-4-9B and

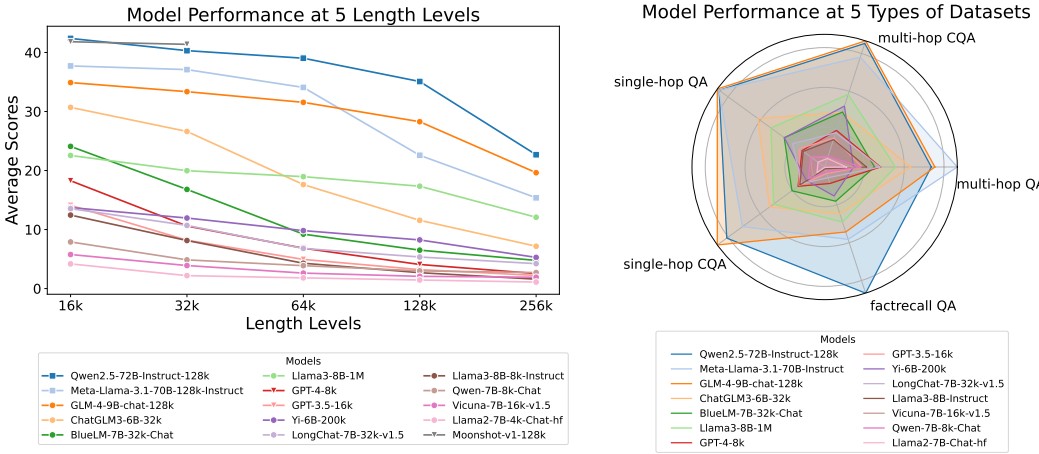

Figure 5: Overall results on different length levels and types of datasets. (a) Average scores across all datasets of 15 LLMs at 5 length levels. The circle markers represent open-source models of 6-9B; The square markers represent larger open-source models around 70B; The triangle markers represent commercial APIs. Note that we only evaluate Moonshot-v1-128k on 16*k* and 32*k* length levels due to the high cost. (b) Average scores across all length levels of 14 LLMs on 5 types of datasets. "CQA" refers to QA datasets with CFI.

Llama-3.1-70B. (iv) Models exhibit a sharp performance drop, often occurring prior to or when the context length exceeds their supported context length. Some models experience a sharp decline in performance once the context length exceeds their supported context length. For instance, Qwen2.5-72B-128k, GLM-4-9B-128k, and ChatGLM3-6B-32k exhibit a sharp performance drop after 128*k*, 128*k*, and 32*k*, respectively. In contrast, some other models encounter this sharp performance drop earlier. For instance, Llama-3.1-70B-128k exhibit a sharp performance drop after 64*k*, despite its stated support for 128*k* context.

Figure 5 (b) shows the average scores across all 5 length levels on 5 task types. We can see that (i) LLMs attain lower scores on multi-hop QA tasks compared to single-hop QA tasks. (ii) Confusing facts insertion adds complexity to the tasks, particularly evident in single-hop QA and single-hop confusion QA. See Appendix E for more detailed results and Appendix H for example failure cases.

## 3.2 Ablation Study of *LV*-Eval Techniques

**Confusing facts insertion.** Table A7, A8, and A9 show the scores of multiple LLMs on dataset with and without CFI. We can see that (i) On **multifieldqa-en-mixup** and **multifieldqa-zh-mixup**, CFI leads to a notable degradation in the scores of LLMs. However, CFI in the **hotpotwikiqa-mixup** dataset does not result in severe degradation. (ii) Table A8 and A9 show that a strong model, ChatGLM3-6B-32k, exhibits the most substantial score degradation on data with CFI. For instance, the score of ChatGLM3-6B-32k degrades from 41.46 to 31.97 (a degradation of 9.49) on the 16*k* length level of **multifieldqa-en-mixup**, while the score degradation of other 5 LLMs falls within the range [0.47, 4.89]. This observation suggests that current powerful LLMs may even be more susceptible to confusing information in the context. Future research is needed to enhance the models' ability to discern information that appears similar but is in fact unrelated. (iii) As the length of the input context increases, the score degradation becomes smaller. This phenomenon can be attributed to two factors: the truncation of confusing facts and a decrease in baseline performance.

**Keyword and phrase replacement.** The technique of KPR aims to eliminate the knowledge leakage and common-sense memorization of LLMs. Intuitively, for datasets sourced from Wikipedia and other widely used corpus, the risk of knowledge leakage is higher. From the results in Table A7, A8, and A9, we observe that: (i) KPR brings notable degradation of

LLM scores on these three datasets suggesting that knowledge leakage exists in open-source corpus and can be mitigated by KPR. (ii) The extent of degradation is relatively consistent across different length levels.

We illustrate the knowledge leakage issue and the impact of KPR in Table A11. Specifically, we compare three settings: (i) Directly querying the LLMs to answer the question without the context ("direct (w.o. KPR)"). (ii) Applying KPR to the QA pair, and directly querying the LLMs without the context ("direct (w. KPR)"). (iii) Applying KPR to the QA pair and the context, and querying the LLMs to answer the question with the context ("w. context (w. KPR)").

Table A11 shows that without KPR, some LLMs can achieve a considerable score even without context. For instance, Yi-6B-200k and ChatGLM3-6B-32k achieve scores of 16.11 and 12.24, respectively, through memorization or common-sense knowledge. Applying KPR decreases the score without context (6.06 for Yi-6B-200k and 4.96 for ChatGLM3-6B-32k). This helps mitigate the influence of memorization or common-sense knowledge on the assessment of long-context understanding ability.

**Case study on the fact-recall tasks.** The **factrecall-en** and **factrecall-zh** datasets are constructed to evaluate the enhanced NIAH (Kamradt, 2023) ability. The traditional NIAH evaluation is basically a retrieval task, asking LLMs to find the answer or passkey in long context, which is too simple for majority of LLMs that they can easily get high scores after task oriented training. Therefore we enhance the NIAH evaluation with CFI and KPR to assess LLM's positional consistency of retrieval while challenging their comprehension and anti-interference abiility.

We show the ablation results of CFI and KPR in Figure 6 and Table A10. From the first column in Figure 6, we can see that ChatGLM3-6B-32k attains high accuracy on datasets without CFI and KPR, as long as the input context length is within its context size (32k). However, when either CFI (second column of sub-figure) or KPR (third column sub-figure) is applied, the retrieval accuracy decreases. The accuracy experiences a more severe degradation when both CFI and KPR are applied, particularly evident in **factrecall-zh**, where a performance collapse is observed (See Appendix H for an example failure case). This indicates that there is room for improvement in the model's ability to accurately identify a specific piece of information from a long context in the presence of interference.

**Keyword-recall-based metric.** For a given length level $L_d$ of the dataset, if the single key information is uniformly distributed in the context, an LLM with a context window size $L_m$ can only observe the key information for approximately $\frac{L_m}{L_d}$ of the time. Thanks to our KPR technique, we can reasonably expect that the LLM cannot get the correct answer through memorization or common-sense knowledge. Furthermore, in our free-form QA tasks, unlike in multiple-choice settings, the LLM cannot easily guess the correct answer. Therefore, we would not expect to see a metric score much higher than $\frac{L_m}{L_d}$. In other words, $\frac{L_m}{L_d}$ is the ideal score assuming that all answers are correctly extracted from and only from the context, and the metric accurately measures the answer correctness.

However, as shown in Table 3, when using the original F1 metric, the metric score can be significantly higher than $\frac{L_m}{L_d}$ due to the undesired matching of non-keywords and non-informative words. For example, ChatGLM3-6B-32k achieves a score of 26.43% on the 256$k$ length level of the **cmrc-mixup** dataset, which greatly exceeds $\frac{L_m}{L_d} = 12.5\%$. In contrast, our keyword-recall-based metric with the word blacklist is a more meaningful metric, as its scores are more aligned with our expectations (i.e., smaller than $\frac{L_m}{L_d}$).

## 4 Limitations and Future Work

*LV*-Eval does not encompass other task types such as summarization. Additionally, due to the high cost, we do not test some of the most recent LLMs, such as GPT-4-128k, GPT-4o, and so on. As we release all the test data, one can intentionally overfit the benchmark by

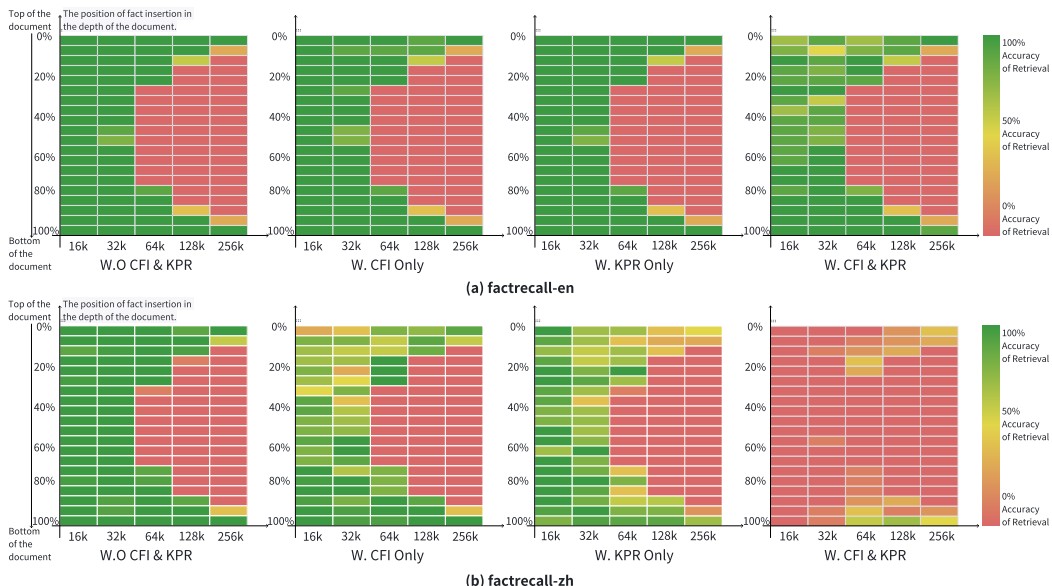

Figure 6: Ablation results of the "needle in a haystack" task on ChatGLM3-6B-32k. (a) **factrecall-en**. (b) **factrecall-zh**. In each of (a)(b), from left to right, the four sub-figures show the results of w.o. "CFI and KPR", "w. CFI only", "w. KPR only", and "w. both CFI and KPR", respectively. These results illustrate that CFI and KPR are effective in improving the task difficulty.

| Metric | 16k | 32k | 64k | 128k | 256k |
|---|---|---|---|---|---|
| reference $\frac{L_m}{L_d}$ | 100 | 100 | 50.00 | 25.00 | 12.50 |
| original | 66.49 | 59.99 | 38.71 | 31.76 | 26.43 |
| w. answer keywords | 57.67 | 52.18 | 28.92 | 21.07 | 15.45 |
| w. answer keywords + word blacklist | 51.21 | 46.34 | 20.71 | 14.16 | 8.38 |

Table 3: Metric scores of ChatGLM3-6B-32k on **cmrc-mixup**. The score inflation is suppressed with keyword-recall-based metric design.

training on the test data to get a high score. In this case, training on LV-Eval datasets with KPR might lead to mistakes in common-sense knowledge, resulting in a very unreliable evaluation.

As *LV*-Eval primarily focuses on extracting and manipulating information from long *inputs*, we do not include questions that require long *outputs*. In future benchmarks where evaluating long outputs is necessary, a more appropriate metric might involve using an LLM rater to assess whether the answer matches the ground truth. However, since our benchmark does not focus on long outputs, we rely on human efforts to provide a simpler metric, with the goal of avoiding the use of a strong LLM in the evaluation process, as this could introduce additional biases and costs.

A worth-noting issue about KPR is that as KPR modifies the fact, there exist some cases where the model identifies factual errors and then insists on providing a common-sense response. We view these cases as an issue with the model's instruction-following ability, as our prompt explicitly states that the answer should be solely based on the context instead of its existing knowledge.

Our CFI technique relies on manual revision of the confusing facts to ensure the benchmark quality. A possible way to alleviate the human efforts when applying the CFI technique to future corpus might be leveraging stronger LLMs to substitute manual revision efforts.

## Acknowledgments

We thank the insightful discussion with Kai Chen and Songyang Zhang from Shanghai Artificial Intelligence Laboratory. Thanks to all of the annotators who contribute to the project. We thank Hanling Zhang for helping run the experiment for Moonshot.

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

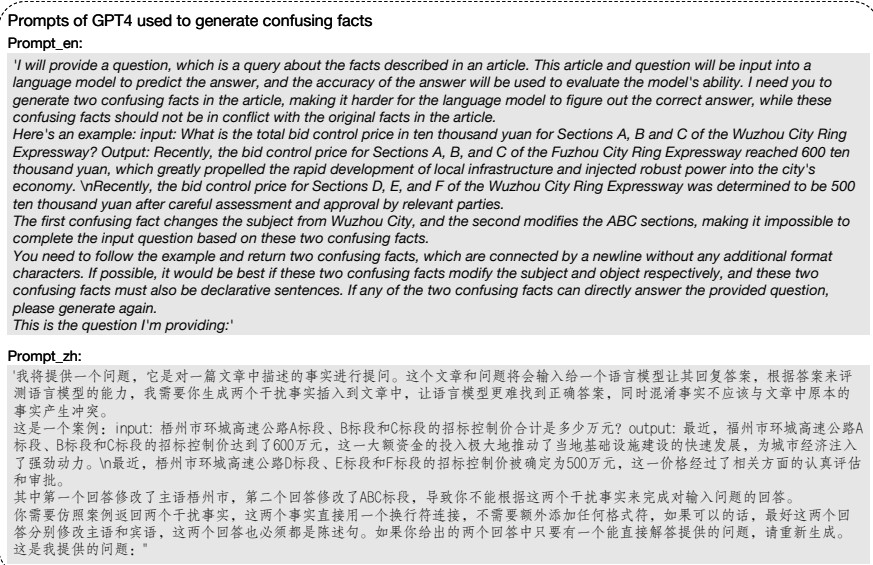

Figure A7: Prompts used for GPT-4 to generate confusing facts.

# A    Related Work

**Long-Context Benchmarks.**    Table 1 provides a summary of existing long-context benchmarks, including ZeroScrolls (Shaham et al., 2023), LooGLE (Li et al., 2023b), L-Eval (An et al., 2023), BAMBOO (Dong et al., 2023), and LongBench (Bai et al., 2023b). ZeroScrolls, LooGLE, and L-Eval are monolingual benchmarks without explicit length level partition. Their average word counts are $\sim14k$, $\sim21k$ and $\sim13.5k$, respectively. In order to evaluate the model's capability across various context lengths, BAMBOO and LongBench have designed various length levels. However, the word counts ($\sim5k$, $\sim9.5k$) of the contexts in these two benchmarks are notably smaller than the supported context length of state-of-the-art long-context models, making them unsuitable for evaluating the claimed extremely long-context understanding ability. In contrast, *LV*-Eval contains five length levels, up to $256k$ words, each with the same set of QA pairs for controllable evaluation.

In terms of metric design, L-Eval introduces a length-instruction-enhanced metric to mitigate the undesired impact of the answer length on metric scores. Additionally, L-Eval proposes to use LLMs to assist in scoring. In *LV*-Eval, we ask human annotators to mark the answer keywords and create a non-informative word blacklist, and propose a two-stage metric to focus more on the answer keywords while reducing the influences of non-informative words.

The "needle in a haystack" (NIAH) task (Kamradt, 2023) has been very popular in assessing the long-context retrieval ability of LLMs. A recent benchmark, RULER (Hsieh et al., 2024), identifies the necessity of extending the NIAH task. It extends NIAH with diverse types and quantities of needles and distracting information. In our work, *LV*-Eval also includes an extended version of the NIAH task with our confusing fact insertion and keyword-and-phrase replacement techniques.

**Long-Context Techniques.**    Considerable efforts have been devoted to enhancing the long-context abilities of LLMs. One line of work focuses on making LLMs have extended context sizes without fine-tuning and behave normally on inputs longer than their training context lengths. The design and extrapolation method of the position encoding module (Su et al., 2024; Press et al., 2021; bloc97, 2023) is crucial for this goal. Besides, several sparse attention techniques (Han et al., 2023; Xiao et al., 2023) have also been proposed to avoid model collapse. These sparse attention techniques also alleviate the quadratic complexity w.r.t. the sequence length.

There are many other strategies aimed at enabling LLMs to effectively leverage long input contexts. The most commonly utilized strategy is long-context fine-tuning (Xiong et al., 2023; Li et al., 2023a; Peng et al., 2023b). For instance, YaRN (Peng et al., 2023b) conducts fine-tuning with 64*k* and 128*k* context lengths starting with Llama2-7B/13B, and Yi-6B-200k (Yi, 2023) is trained with 200*k* context length starting with its 4*k* variant. Other strategies include the recurrent- or memory-based architecture (Dai et al., 2019; Wu et al., 2021; Martins et al., 2022; Zhou et al., 2023b; Liang et al., 2023), and the retrieval- or summarization-based context compression techniques (Khandelwal et al., 2019; Borgeaud et al., 2022; Bai et al., 2023b; Zhou et al., 2023b), and so on.

In this work, we evaluate LLMs of diverse context sizes, ranging from 4*k* to 200*k*, most of which have incorporated advanced position encoding design and undergone long-context fine-tuning.

## B  Discussion

**Could the CF generated by GPT-4 give an unfair effect to the evaluation results of GPT-4?**
There is a possibility that the CF generated by GPT-4 could lead to either an unfair advantage or an unfair disadvantage in its evaluation results. On the one hand, the generated CF is aligned to GPT-4's internal concept, which may lead to an unfair advantage, due to the problem of data leakage. On the other hand, it may lead to an unfair disadvantage, because the CF generated by GPT-4 itself may be the ones that are hardest for GPT-4 to distinguish, unless explicitly prompted to consider CF within the context.

**Comparison of LV-Eval with two concurrently developed benchmarks: XLBench (Ni et al., 2024) and Loong (Wang et al., 2024)**    There are several key differences between LV-Eval and these two benchmarks: 1. Context Length Balance: Both Loong and XLBench are not length-balanced, meaning that test instances of different lengths do not share the same QA pairs. In contrast, LV-Eval ensures that the supporting documents and QA pairs are shared across different length levels, making it easier to track performance degradation as context length increases. 2. Application Focus: Loong is primarily focused on information extraction tasks, such as sheet analysis, which limits its scope. In contrast, LV-Eval is a more general-purpose benchmark, assessing not only knowledge extraction but also manipulation and understanding in complex long-context scenarios. 3. Benchmarking Techniques: We introduce the KPR benchmarking technique to decouple the evaluation of common-sense knowledge from the evaluation of long-context understanding, addressing a key limitation in existing long-context benchmarks. Additionally, we propose an optimized metric, leveraging manual efforts to label keywords, which makes the scoring more intuitive and interpretable (see Table 3). We hope this comparison clarifies the distinct contributions of LV-Eval to the community.

## C  Detailed Construction of QA Pairs

As LV-Eval primarily focuses on extracting and manipulating information from long inputs, we do not include questions that require long outputs. We select QA pairs with short and brief answers.

**Multi-hop QA.**    In a multi-hop QA task, the reasoning to derive the answer needs to gather multiple pieces of information from various locations in the context. We construct four multi-hop QA datasets: **dureader-mixup**, **loogle-CR-mixup**, **loogle-MR-mixup**, and **hotpotwikiqa-mixup**.

- **hotpotwikiqa-mixup** is originated from two Wikipedia-based multi-hop QA datasets: HotpotQA and 2WikiMultihopQA. HotpotQA contains 112,779 2-hop questions that are written by native speakers according to two given paragraphs as the context. 2WikiMultihopQA contains 192,606 5-hop questions that are synthesized using manually designed templates to prevent shortcut solutions. We select 124 samples from the two datasets.

- **loogle-MR-mixup** and **loogle-CR-mixup** originate from LooGLE's Long-dependency QA task, specifically the *Multiple information Retrieval* and *Comprehension and Reasoning* subtasks. The *Multiple information Retrieval* task requires aggregation of the evidence that can be directly located in original sentences, while the *Comprehension and Reasoning* task contains implicit evidence within the context, it requires multi-step reasoning to get the correct answers. We select 139 and 99 questions for **loogle-MR-mixup** and **loogle-CR-mixup**, respectively.

- **dureader-mixup** is built from the DuReader dataset. We first randomly select 200 instances and then manually remove 24 samples whose answers are longer than 360 words.

**Single-hop QA.** In a single-hop QA task, only a single evidence in the context is needed to derive the answer. We construct seven single-hop QA datasets: **lic-mixup**, **loogle-SD-mixup**, **cmrc-mixup**, **multifieldqa-en-mixup**, **multifieldqa-zh-mixup**, **factrecall-en**, and **factrecall-zh**.

- **lic-mixup** is originated from the Long-instruction-en2zh dataset on Hugging Face. Long-instruction-en2zh contains 8,000+ high-quality Chinese multi-doc QA data translated from English. We selected 197 QA pairs and their corresponding documents as supporting data, while the remaining documents serve as distracting data for context mixing.

- **loogle-SD-mixup** contains 160 unique QA pairs and 800 documents originated from the short-dependency QA task in LooGLE.

- **cmrc-mixup** is derived from the CMRC 2018 Public Datasets, designed for Chinese machine reading comprehension. It contains $\sim 20k$ questions annotated on Wikipedia documents by human experts. We manually pick 200 QA pairs and their corresponding documents as supporting QA pairs and documents.

- **multifieldqa-en-mixup** and **multifieldqa-zh-mixup** are built from the Multi-FieldQA datasets in Long-Bench. We manually remove questions that can be answered using common-sense knowledge without referring to the context, and eventually get 101 and 133 unique QA pairs for **multifieldqa-en-mixup** and **multifieldqa-zh-mixup**, respectively.

- **factrecall-en** and **factrecall-zh** are two synthetic datasets designed to assess the LLMs' ability to identify a small piece of evidence ("fact") located at various locations within a lengthy context. As shown in Figure A12 A13, we write one English fact-question-answer pair for **factrecall-en** and one Chinese fact-question-answer pair for **factrecall-zh**. distracting documents are sourced from *PG-19* dataset (English) and the book of *Journey to the West* (Chinese) to create five contexts of different length levels. For each context, we generate 200 documents by inserting the fact at 200 evenly spaced positions within the context.

## D  Annotation Details

- **Annotators**: We hire a total of five annotators, including three master students involved in LLM research and two master students majored in linguistics. They are hired to work on-site as full-time annotators.

- **Annotation time**: On average, annotators worked 8 hours per day. (1) For CFI, six datasets comprising a total of 557 instances of confusing facts were reviewed, with the verification process completed by two annotators over a period of three days. This task was assigned to master's students in linguistics to ensure semantic consistency. (2) For KPR, six datasets containing 1,924 pairs were processed. Replacing key words and phrases in the Chinese datasets involved five individuals over three days, whereas the English datasets required two individuals over two days. (3) For answer keyword annotation, nine datasets comprising 955 instances of answer keywords were annotated by two individuals within a single day.

| Model Name | SFT | Context Length | HuggingFace / API Endpoint |
|---|---|---|---|
| Llama2-7B-Chat-hf (Touvron et al., 2023) | ✓ | 4k | meta-llama/Llama-2-7b-chat-hf |
| Qwen-7B-8k-Chat (Bai et al., 2023a) | ✓ | 8k | Qwen/Qwen-7B-Chat |
| Llama3-8B-Instruct (AI@Meta, 2024) | ✓ | 8k | meta-llama/Meta-Llama-3-8B-Instruct |
| Vicuna-7B-16k-v1.5 (Zheng et al., 2023) | ✓ | 16k | lmsys/vicuna-7b-v1.5-16k |
| ChatGLM3-6B-32k (Zeng et al., 2022) | ✓ | 32k | THUDM/chatglm3-6b-32k |
| BlueLM-7B-32k-Chat (Team, 2023) | ✓ | 32k | vivo-ai/BlueLM-7B-Chat-32K |
| LongChat-7B-32k-v1.5 (togetherAI, 2023) | ✓ | 32k | lmsys/longchat-7b-v1.5-32k |
| GLM-4-9B-128k (GLM, 2024) | ✓ | 128k | THUDM/glm-4-9b-chat |
| Qwen2.5-72B-Instruct-128k (Yang et al., 2024) | ✓ | 128k | Qwen/Qwen2.5-72B-Instruct |
| Meta-Llama-3.1-70B-128k-Instruct (Meta, 2024) | ✓ | 128k | meta-llama/Llama-3.1-70B-Instruct |
| Yi-6B-200k (Yi, 2023) | | 200k | 01-ai/Yi-6B-200K |
| Llama3-8B-1M (gradient.ai, 2024) | ✓ | 1048k | gradientai/Llama-3-8B-Instruct-Gradient-1048k |
| GPT-4-8k (Achiam et al., 2023) | ✓ | 8k | gpt-4-0613 |
| GPT-3.5-16k (Ye et al., 2023) | ✓ | 16k | gpt-3.5-turbo-1106 |
| Moonshot-V1-128k (moonshot, 2024) | ✓ | 128k | moonshot-V1-128k |

Table A4: Information of evaluated LLMs.

- **Annotation guidelines**: We provided guidelines and examples to the five annotators and conducted a trial annotation in accordance with these guidelines to ensure they understand the guidelines. The guidelines are summarized as follows. For CFI, two types of cases require manual modification: (a) The confusing facts generated by GPT-4 fail to produce interference; (b) The confusing facts generated by GPT-4 conflict with the original facts.

  For KPR, the guidelines are: (a) Ensure that the replaced words or phrases are distinct from the original and not synonyms; (b) Maximize the differences between the revised and original sentences by replacing as many words as possible; (c) Prioritize the replacement of words that do not have synonyms, as it is challenging to ensure that all synonyms are correctly replaced for words with multiple synonyms; (d) After replacement, the resulting statement may be inconsistent with common knowledge; (e) After replacement, it's OK that the resulting statement may be inconsistent with common knowledge, as long as the answer can be derived from the context and that all information related to this answer is consistent within the context.

- **Measurement of inter-annotator agreement**: We did not measure inter-annotator agreement during the annotation process. Since the principles of the annotation task are simple and clear, and the number of annotations is relatively small, we had the two annotators review each other's work to resolve any disagreements after completing their independent annotations.

## E  Detailed Evaluation Results

The detailed information of evaluated LLMs are shown in Table A4.

The detailed results on each dataset of the single-hop QA task type and multi-hop QA task type are shown in Figure A8 and Figure A9, respectively. We can see that (i) Among the multi-hop QA datasets, **loogle-CR-mixup** and **loogle-MR-mixup** are particularly challenging. Future research is needed to improve the ability to aggregate multiple pieces of evidence from a long context with distracting and confusing facts. (ii) For single-hop QA datasets, as expected, LLMs can achieve higher scores on datasets without CFI, including **loogle-SD-mixup** and **cmrc-mixup**. (iii) Several LLMs, namely ChatGLM3-6B-32k, BlueLM-7B-32k-Chat, and Yi-6B-200k, can achieve relatively high scores on **factrecall-en**. This indicates that the NIAH task might not be challenging enough, emphasizing the need to evaluate LLMs on other tasks, particularly multi-hop QA datasets. (iv) The performance gap between LLMs on **factrecall-en** and **factrecall-zh** is especially large, and some open-source LLMs with relatively small context sizes, namely Llama2-7B-Chat-hf (4k context window size), Qwen-7B-8k-Chat, and Vicuna-7B-16k-v1.5, even get near-zero scores. (v) A few LLMs have unbalanced performances on Chinese and English datasets, as illustrated by the results on **multifieldqa-en-mixup** and **multifieldqa-zh-mixup**. The detailed scores of all models on 5 length levels of all sub-datasets are shown in Table A5 A6.

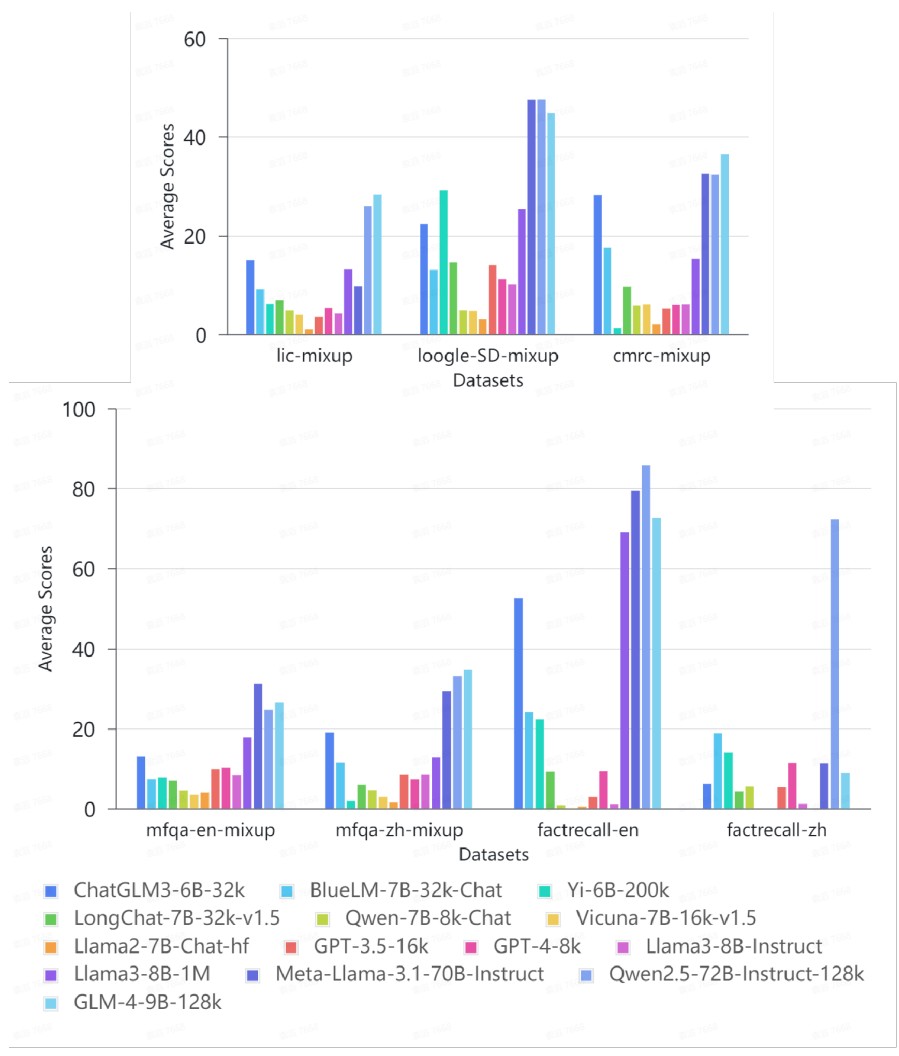

Figure A8: Average scores across all length levels of 14 LLMs on single-hop QA datasets.

Additionally, we select the top 3 LLMs from our study for evaluation with LLM auto rater. We select GPT-4-0613 as auto rater, and the auto rating prompts, which is shown in Figure A10, are well-crafted to make evaluation objective. The average scores on five length levels of those 3 LLMs are plot in dot lines in Figure A11. The results suggest that auto rater generally assigns higher scores than our keyword-recall-based metric. Despite this, the relative model performance is still accurately reflected. LLM auto rater consumes about 2M input tokens in the case of taking GPT-4-0613 as judge model, which incurs an approximate cost of 20 $ per model evaluation.

# F Detailed Ablation Results

The detailed ablation results of CFI and KPR are shown in Table A7 A8 A9.

# G Samples in *LV*-Eval

For completeness of our manuscript, we show some data samples of **factrecall-en** and **factrecall-zh** in Figure A12 A13.

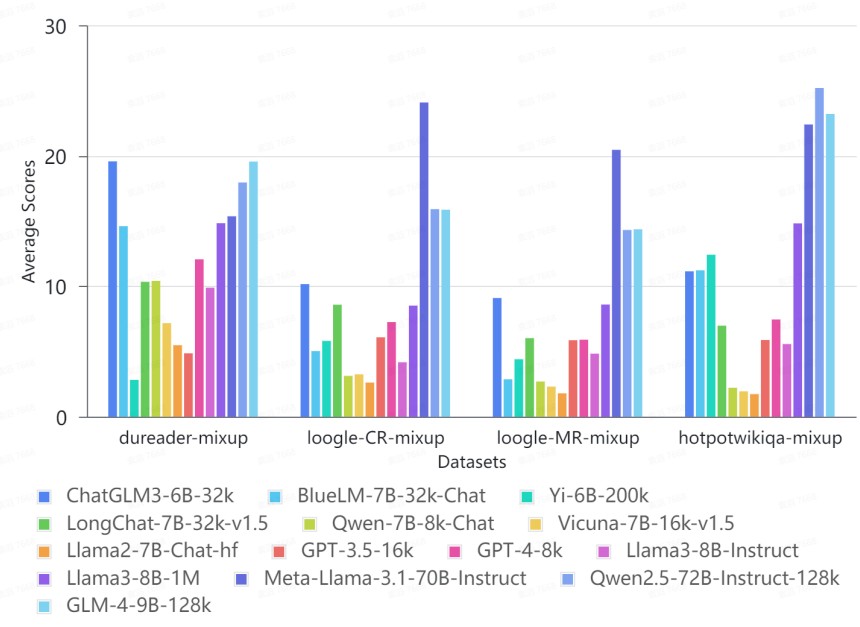

Figure A9: Average scores across all length levels of 14 LLMs on multi-hop QA datasets.

| Dataset | Len. | Ch. | Bl. | Yi. | Lo. | Qw. | Vi. | Ll.4 | Ll.8 | Ll.M | GPT3. | GPT4. | Ll.31 | Qw.25 | GLM9. | Ms. |
|---|---|---|---|---|---|---|---|---|---|---|---|---|---|---|---|---|
| **dureader -mixup** | 16*k* | 23.99 | 19.40 | 2.87 | 13.44 | 11.82 | 9.67 | 7.21 | 16.39 | 18.06 | 8.01 | 19.14 | 20.22 | 18.28 | 19.71 | 21.46 |
| | 32*k* | 25.21 | 19.74 | 2.98 | 11.57 | 12.80 | 7.65 | 5.42 | 13.08 | 15.86 | 5.26 | 13.64 | 22.38 | 19.16 | 19.15 | 21.00 |
| | 64*k* | 22.01 | 14.44 | 2.88 | 9.23 | 10.48 | 6.62 | 5.59 | 10.24 | 15.16 | 4.26 | 12.66 | 19.22 | 17.82 | 21.02 | - |
| | 128*k* | 17.94 | 10.95 | 2.36 | 9.51 | 8.15 | 6.25 | 4.78 | 5.30 | 14.46 | 3.30 | 8.19 | 9.90 | 18.24 | 19.37 | - |
| | 256*k* | 8.72 | 8.51 | 3.06 | 7.96 | 8.65 | 5.70 | 4.45 | 4.46 | 10.64 | 3.50 | 6.71 | 5.11 | 16.27 | 18.54 | - |
| **loogle-CR -mixup** | 16*k* | 14.41 | 9.01 | 8.25 | 11.25 | 5.48 | 5.00 | 3.69 | 8.63 | 12.56 | 10.04 | 12.68 | 30.05 | 20.33 | 21.59 | 25.02 |
| | 32*k* | 14.10 | 7.36 | 8.83 | 11.17 | 3.30 | 4.25 | 3.29 | 8.74 | 11.05 | 8.39 | 10.40 | 26.24 | 20.25 | 20.53 | 23.29 |
| | 64*k* | 9.92 | 3.81 | 4.73 | 9.31 | 3.82 | 3.76 | 3.13 | 2.78 | 8.64 | 5.58 | 6.48 | 26.81 | 19.08 | 15.55 | - |
| | 128*k* | 6.95 | 2.40 | 4.05 | 6.19 | 1.14 | 1.99 | 2.19 | 0.26 | 5.81 | 3.08 | 2.83 | 20.72 | 12.56 | 14.07 | - |
| | 256*k* | 5.46 | 2.60 | 3.23 | 5.03 | 1.94 | 1.28 | 0.81 | 0.49 | 4.54 | 3.37 | 3.91 | 16.61 | 7.31 | 7.61 | - |
| **loogle-MR -mixup** | 16*k* | 15.83 | 4.90 | 6.94 | 10.53 | 4.93 | 5.17 | 3.37 | 10.39 | 13.73 | 12.95 | 12.24 | 23.60 | 19.14 | 15.88 | 20.62 |
| | 32*k* | 11.62 | 3.14 | 7.67 | 9.51 | 2.95 | 3.83 | 2.20 | 7.14 | 10.9 | 7.03 | 7.83 | 22.54 | 16.12 | 16.29 | 19.63 |
| | 64*k* | 7.00 | 1.68 | 2.69 | 3.04 | 2.37 | 0.96 | 2.05 | 3.89 | 7.82 | 6.23 | 6.26 | 21.57 | 16.52 | 16.41 | - |
| | 128*k* | 7.24 | 2.46 | 3.44 | 4.05 | 1.80 | 0.55 | 1.04 | 2.37 | 5.93 | 2.13 | 2.30 | 18.75 | 12.06 | 11.87 | - |
| | 256*k* | 3.82 | 2.19 | 1.32 | 3.01 | 1.46 | 1.06 | 0.33 | 0.4 | 4.63 | 1.00 | 0.90 | 15.83 | 7.74 | 11.40 | - |
| **hotpotwikiqa -mixup** | 16*k* | 16.98 | 19.31 | 23.55 | 11.57 | 2.78 | 2.63 | 3.99 | 12.14 | 17.67 | 11.96 | 13.51 | 27.53 | 31.69 | 27.60 | 30.08 |
| | 32*k* | 14.76 | 14.07 | 18.94 | 10.71 | 1.89 | 2.19 | 1.30 | 7.37 | 17.17 | 6.66 | 10.62 | 30.86 | 31.96 | 28.98 | 28.93 |
| | 64*k* | 9.02 | 9.63 | 9.94 | 4.77 | 2.27 | 2.05 | 1.84 | 2.34 | 13.37 | 3.27 | 6.67 | 27.57 | 26.80 | 24.42 | - |
| | 128*k* | 8.31 | 7.71 | 7.66 | 5.49 | 2.37 | 1.04 | 0.81 | 3.86 | 15.02 | 4.23 | 4.13 | 17.97 | 21.46 | 21.06 | - |
| | 256*k* | 6.68 | 5.40 | 2.01 | 2.37 | 1.82 | 1.85 | 0.75 | 2.17 | 10.88 | 3.30 | 2.36 | 8.07 | 14.07 | 13.98 | - |

Table A5: Overall results of multi-hop QA tasks in *LV*-Eval. The abbreviations "Ch.", "Bl.", "Yi.", "Lo.", "Qw.", "Vi.", "Ll.4", "Ll.8", "Ll.M", "Ll.31", "Qw.25", "GLM9.", "Ms.", "GPT3.", and "GPT4." stand for ChatGLM3-6B-32k, BlueLM-7B-32k-Chat, Yi-6B-200k, LongChat-7B-32k-v1.5, Qwen-7B-8k-Chat, Vicuna-7B-16k-v1.5, Llama2-7B-Chat-hf, Llama3-8B-Instruct, Llama3-8B-1M, Meta-Llama-3.1-70B-Instruct, Qwen2.5-72B-Instruct-128k, GLM-4-9B-128k, moonshot-v1-128k, GPT-3.5-16k, and GPT-4-8k, respectively.

# H   Examples of Failure Cases

Figure A14 shows some failure cases of Llama-3-8b-Instruct when confusing facts exist in **factrecall-en** and **factrecall-zh**. In **factrecall-zh-16k**, all of Llama-3-8b-Instruct's responses were misled by the confusing fact, that is "David Beckham", whereas in the **factrecall-en-16k** dataset, only 32% of the responses were misled by the confusing fact. This suggests that the model's anti-interference ability may vary significantly across languages.

Figure A15 shows a failure case of multi-hop reasoning in a QA task. In the 65th test sample of the **hotpotwikiqa-mixup-16k** dataset, the question asks, `What is the date of death of the director of the film Nallavan Vazhvan?`. An-

| Dataset | Len. | Ch. | Bl. | Yi. | Lo. | Qw. | Vi. | Ll.4 | Ll.8 | Ll.M | GPT3. | GPT4. | Ll.31 | Qw.25 | GLM9. | Ms. |
|---|---|---|---|---|---|---|---|---|---|---|---|---|---|---|---|---|
| lic-mixup | 16k | 24.15 | 20.75 | 5.37 | 15.45 | 6.05 | 8.34 | 2.48 | 9.16 | 15.27 | 7.65 | 13.69 | 15.46 | 30.96 | 31.11 | 35.57 |
| | 32k | 22.27 | 12.68 | 6.25 | 10.02 | 6.07 | 4.81 | 0.99 | 6.57 | 15.62 | 4.42 | 5.86 | 17.02 | 28.14 | 30.00 | 34.46 |
| | 64k | 14.33 | 5.00 | 7.19 | 4.54 | 4.21 | 2.52 | 0.48 | 1.80 | 14.85 | 3.07 | 3.23 | 12.77 | 27.65 | 31.84 | - |
| | 128k | 8.30 | 3.03 | 5.56 | 2.47 | 4.34 | 2.36 | 0.42 | 1.30 | 10.35 | 0.87 | 1.90 | 2.17 | 27.71 | 30.35 | - |
| | 256k | 6.07 | 4.11 | 6.24 | 2.14 | 3.19 | 1.99 | 0.73 | 1.30 | 7.96 | 1.65 | 1.70 | 1.32 | 15.20 | 17.99 | - |
| loogle-SD -mixup | 16k | 41.82 | 34.34 | 39.56 | 27.42 | 10.54 | 8.79 | 6.75 | 25.08 | 39.53 | 31.67 | 27.01 | 63.25 | 58.13 | 57.23 | 59.83 |
| | 32k | 30.31 | 15.10 | 36.48 | 18.21 | 4.70 | 4.90 | 2.61 | 12.56 | 31.45 | 18.56 | 14.01 | 61.95 | 57.02 | 52.98 | 57.85 |
| | 64k | 19.07 | 4.95 | 31.71 | 12.09 | 2.40 | 3.07 | 2.58 | 7.34 | 28.45 | 10.41 | 8.00 | 53.97 | 56.38 | 47.83 | - |
| | 128k | 11.34 | 5.32 | 25.71 | 9.11 | 3.25 | 4.24 | 2.04 | 4.85 | 18.81 | 5.74 | 5.14 | 35.63 | 41.99 | 43.15 | - |
| | 256k | 8.92 | 5.41 | 12.37 | 5.97 | 3.02 | 2.39 | 1.24 | 0.91 | 8.37 | 1.48 | | 22.45 | 23.95 | 22.62 | - |
| cmrc-mixup | 16k | 51.21 | 45.89 | 1.05 | 20.99 | 11.13 | 11.75 | 3.85 | 15.16 | 20.25 | 12.19 | 14.67 | 53.95 | 35.63 | 42.42 | 46.42 |
| | 32k | 46.34 | 19.53 | 0.35 | 10.77 | 5.32 | 6.55 | 1.08 | 6.77 | 19.83 | 6.00 | 3.33 | 49.82 | 33.01 | 42.01 | 46.80 |
| | 64k | 20.71 | 10.66 | 0.84 | 8.97 | 4.68 | 5.04 | 1.72 | 4.82 | 17.27 | 3.57 | 5.31 | 50.72 | 35.38 | 38.50 | - |
| | 128k | 14.16 | 7.06 | 1.58 | 3.77 | 3.81 | 2.75 | 1.64 | 1.78 | 13.46 | 2.73 | 3.81 | 5.32 | 35.22 | 36.67 | - |
| | 256k | 8.38 | 4.51 | 2.54 | 3.75 | 4.09 | 4.13 | 1.54 | 1.73 | 5.66 | 1.32 | 2.68 | 2.94 | 22.38 | 22.56 | - |
| multifieldqa -en-mixup | 16k | 25.40 | 11.82 | 10.01 | 12.02 | 7.66 | 6.29 | 8.81 | 16.33 | 21.30 | 18.78 | 19.00 | 39.46 | 29.90 | 32.06 | 33.44 |
| | 32k | 12.78 | 6.34 | 9.24 | 7.58 | 3.61 | 4.32 | 5.55 | 9.60 | 17.05 | 11.59 | 12.69 | 38.61 | 27.96 | 31.18 | 34.19 |
| | 64k | 12.32 | 8.38 | 8.83 | 7.84 | 5.23 | 2.79 | 1.58 | 6.15 | 18.68 | 7.38 | 8.30 | 32.19 | 26.87 | 27.89 | - |
| | 128k | 9.89 | 5.29 | 5.98 | 3.11 | 3.64 | 2.51 | 2.54 | 6.63 | 17.27 | 7.95 | 7.25 | 26.06 | 21.21 | 24.25 | - |
| | 256k | 4.24 | 4.78 | 4.69 | 4.22 | 2.44 | 1.28 | 1.49 | 3.20 | 3.21 | 3.54 | | 19.05 | 17.10 | 17.27 | - |
| multifieldqa -zh-mixup | 16k | 32.38 | 22.05 | 2.85 | 9.81 | 8.82 | 5.82 | 4.72 | 18.73 | 21.69 | 18.94 | 17.61 | 39.16 | 40.65 | 39.44 | 39.05 |
| | 32k | 24.48 | 17.64 | 0.75 | 8.82 | 5.68 | 4.45 | 1.21 | 13.60 | 13.46 | 12.21 | 11.18 | 32.80 | 36.28 | 35.84 | 31.71 |
| | 64k | 20.97 | 7.36 | 1.89 | 3.23 | 3.01 | 2.03 | 0.68 | 6.13 | 11.31 | 6.29 | 4.99 | 30.77 | 31.79 | 36.13 | - |
| | 128k | 10.08 | 5.90 | 2.11 | 3.54 | 2.84 | 0.88 | 0.24 | 1.52 | 9.28 | 2.94 | 1.76 | 24.42 | 30.71 | 35.13 | - |
| | 256k | 7.05 | 4.48 | 1.58 | 3.92 | 2.52 | 1.26 | 0.56 | 2.62 | 7.79 | 2.15 | 0.92 | 19.37 | 25.95 | 26.42 | - |
| factrecall-en | 16k | 91.50 | 58.5 | 24.88 | 9.22 | 1.77 | 0 | 1.08 | 2.72 | 68.00 | 8.25 | 23.4 | 85.20 | 98.50 | 86.35 | 81.38 |
| | 32k | 89.00 | 32.17 | 23.09 | 14.33 | 1.12 | 0 | 0.46 | 3.27 | 67.17 | 3.27 | 11.84 | 88.36 | 96.00 | 82.00 | 86.17 |
| | 64k | 46.00 | 15.50 | 24.96 | 8.31 | 0.71 | 0 | 0.31 | 0.61 | 73.00 | 1.80 | 5.21 | 93.33 | 94.00 | 79.00 | - |
| | 128k | 24.00 | 9.00 | 22.04 | 7.86 | 0.18 | 0.25 | 0.23 | 0.15 | 78.83 | 0.60 | 4.03 | 81.88 | 88.00 | 69.00 | - |
| | 256k | 12.50 | 5.00 | 16.44 | 6.00 | 0.20 | 0.20 | 0.15 | 0 | 58.00 | 0.45 | 1.79 | 47.96 | 52.50 | 46.50 | - |
| factrecall-zh | 16k | 0 | 19.00 | 25.73 | 7.20 | 15.75 | 0 | 0 | 2.18 | 0 | 14.51 | 28.03 | 17.00 | 83.00 | 10.50 | 67.19 |
| | 32k | 2.00 | 37.00 | 16.86 | 5.00 | 6.00 | 0 | 0 | 2.03 | 0.14 | 6.70 | 15.24 | 17.50 | 77.50 | 8.00 | 71.14 |
| | 64k | 12.50 | 20.00 | 12.41 | 3.50 | 3.50 | 0 | 0 | 1.09 | 0 | 2.49 | 8.08 | 6.00 | 77.00 | 8.50 | - |
| | 128k | 9.00 | 12.50 | 10.13 | 3.70 | 1.50 | 0 | 0 | 0.32 | 0 | 1.72 | 3.58 | 5.53 | 76.50 | 6.00 | - |
| | 256k | 7.00 | 5.50 | 4.62 | 2.00 | 0.50 | 0 | 0 | 0.21 | 0 | 0.98 | 2.00 | 10.44 | 47.06 | 11.00 | - |

Table A6: Overall results of single-hop QA tasks in *LV*-Eval. The abbreviations "Ch.", "Bl.", "Yi.", "Lo.", "Qw.", "Vi.", "Ll.4", "Ll.8", "Ll.M", "Ll.31", "Qw.25", "GLM9.", "Ms.", "GPT3.", and "GPT4." stand for ChatGLM3-6B-32k, BlueLM-7B-32k-Chat, Yi-6B-200k, LongChat-7B-32k-v1.5, Qwen-7B-8k-Chat, Vicuna-7B-16k-v1.5, Llama2-7B-Chat-hf, Llama3-8B-Instruct, Llama3-8B-1M, Meta-Llama-3.1-70B-Instruct, Qwen2.5-72B-Instruct-128k, GLM-4-9B-128k, moonshot-v1-128k, GPT-3.5-16k, and GPT-4-8k, respectively.

> Rating prompt for LLM auto judge
>
> For factrecall datasets:
>
> '''*You are a LLM evaluator. There are a refenrece {{answer}}, and a LLM's {{response}}. Your task is to rate the {{response}} based on the following two aspects:*
> *## Semantic similarity between {{answer}} and {{response}}*
> *## {{response}} recall of the {{answer}}*
> *You must rate the response on a scale of 0 to 10 by strictly following this format: "[[rating]]", for example: "[[5]]".*
> *<The start of {{answer}}>*
> *{A}*
> *<The end of {{answer}}>*
> *<The start of {{response}}>*
> *{R}*
> *<The end of {{response}}>'''*
>
> For other datasets:
>
> '''*You are a LLM evaluator. There are a pair of {{question}} and {{answer}}, and a {{response}} from the evaluated LLM. Your task is to rate LLM's {{response}} based on the following two aspects:*
> *## Semantic similarity between {{answer}} and {{response}}*
> *## Is the {{response}} relevant to the {{question}}?*
> *You must rate the response on a scale of 0 to 10 by strictly following this format: "[[rating]]", for example: "[[5]]".*
> *<The start of {{question}}>*
> *{Q}*
> *<The end of {{question}}>*
> *<The start of {{answer}}>*
> *{A}*
> *<The end of {{answer}}>*
> *<The start of {{response}}>*
> *{R}*
> *<The end of {{response}}>'''*

Figure A10: Rating prompt for GPT4 auto judge.

swering this question requires multi-hop reasoning: The model needs to first extract the director's name from ### Passage 30, which discusses the film Nallavan Vazhvan, and then retrieve the final answer from ### Passage 15 which provides biographical details. However, the model incorrectly pulls the answer from ### Passage 27, which describes

| Model Name | Ablation | hotpotwikiqa-mixup | | | | |
|---|---|---|---|---|---|---|
| | | 16k | 32k | 64k | 128k | 256k |
| Llama2-7B-Chat-hf | w. both | 3.99 | 1.30 | 1.84 | 0.81 | 0.75 |
| | w. KPR | 4.10 | 1.56 | 1.36 | 0.63 | 0.88 |
| | w. CFI | 6.29 | 2.47 | 3.37 | 1.47 | 1.57 |
| | w.o. both | 6.48 | 2.48 | 2.98 | 1.29 | 1.57 |
| ChatGLM3-6B-32k | w. both | 16.98 | 14.76 | 9.02 | 8.31 | 6.68 |
| | w. KPR | 21.32 | 13.04 | 9.99 | 6.56 | 6.12 |
| | w. CFI | 27.06 | 24.75 | 17.57 | 12.89 | 10.88 |
| | w.o. both | 28.48 | 21.96 | 18.89 | 11.31 | 10.69 |
| LongChat-7B-32k-v1.5 | w. both | 11.57 | 10.71 | 4.77 | 5.49 | 2.37 |
| | w. KPR | 11.07 | 6.17 | 5.27 | 5.31 | 3.06 |
| | w. CFI | 19.48 | 14.33 | 9.41 | 11.34 | 6.44 |
| | w.o. both | 18.79 | 12.44 | 9.94 | 11.33 | 7.47 |
| Yi-6B-200k | w. both | 23.55 | 18.94 | 9.94 | 7.66 | 2.01 |
| | w. KPR | 23.84 | 13.77 | 6.52 | 6.69 | 3.84 |
| | w. CFI | 33.32 | 16.89 | 11.00 | 7.62 | 8.09 |
| | w.o. both | 30.71 | 17.62 | 10.43 | 10.17 | 8.51 |
| Vicuna-7B-16k-v1.5 | w. both | 2.63 | 2.19 | 2.05 | 1.04 | 1.85 |
| | w. KPR | 2.09 | 1.63 | 1.27 | 1.13 | 1.98 |
| | w. CFI | 5.84 | 3.58 | 2.60 | 1.82 | 1.09 |
| | w.o. both | 5.81 | 4.09 | 3.30 | 1.48 | 1.22 |

Table A7: Ablation results on **hotpotwikiqa-mixup** for confusing facts insertion (CFI) and keyword and phrase replacement (KPR).

| Model Name | Ablation | multifieldqa-en-mixup | | | | |
|---|---|---|---|---|---|---|
| | | 16k | 32k | 64k | 128k | 256k |
| Llama2-7B-Chat-hf | w. both | 8.81 | 5.55 | 1.58 | 2.54 | 1.49 |
| | w. KPR | 8.43 | 4.84 | 1.93 | 2.46 | 0.95 |
| | w. CFI | 9.05 | 6.08 | 3.29 | 3.59 | 1.44 |
| | w.o. both | 9.65 | 6.08 | 3.29 | 3.59 | 1.67 |
| ChatGLM3-6B-32k | w. both | 25.40 | 12.78 | 12.32 | 9.89 | 4.24 |
| | w. KPR | 33.54 | 17.27 | 12.15 | 8.94 | 4.44 |
| | w. CFI | 31.97 | 19.80 | 14.12 | 10.54 | 6.40 |
| | w.o. both | 41.46 | 24.29 | 14.32 | 10.31 | 6.24 |
| LongChat-7B-32k-v1.5 | w. both | 12.02 | 7.58 | 7.84 | 3.11 | 4.22 |
| | w. KPR | 15.32 | 10.61 | 6.49 | 3.02 | 4.94 |
| | w. CFI | 15.56 | 8.77 | 13.16 | 9.88 | 8.65 |
| | w.o. both | 20.45 | 12.91 | 11.69 | 9.28 | 8.59 |
| Yi-6B-200k | w. both | 10.01 | 9.24 | 8.83 | 5.98 | 4.69 |
| | w. KPR | 12.69 | 13.67 | 11.05 | 7.30 | 5.70 |
| | w. CFI | 12.02 | 9.70 | 11.19 | 5.91 | 7.29 |
| | w.o. both | 16.78 | 13.35 | 12.38 | 7.83 | 7.27 |
| Vicuna-7B-16k-v1.5 | w. both | 6.29 | 4.32 | 2.79 | 2.51 | 1.28 |
| | w. KPR | 8.07 | 4.32 | 2.67 | 2.65 | 1.31 |
| | w. CFI | 9.02 | 6.66 | 5.40 | 2.94 | 2.37 |
| | w.o. both | 9.49 | 6.88 | 5.52 | 2.90 | 2.09 |

Table A8: Ablation results on **multifieldqa-en-mixup** for confusing facts insertion (CFI) and keyword and phrase replacement (KPR).

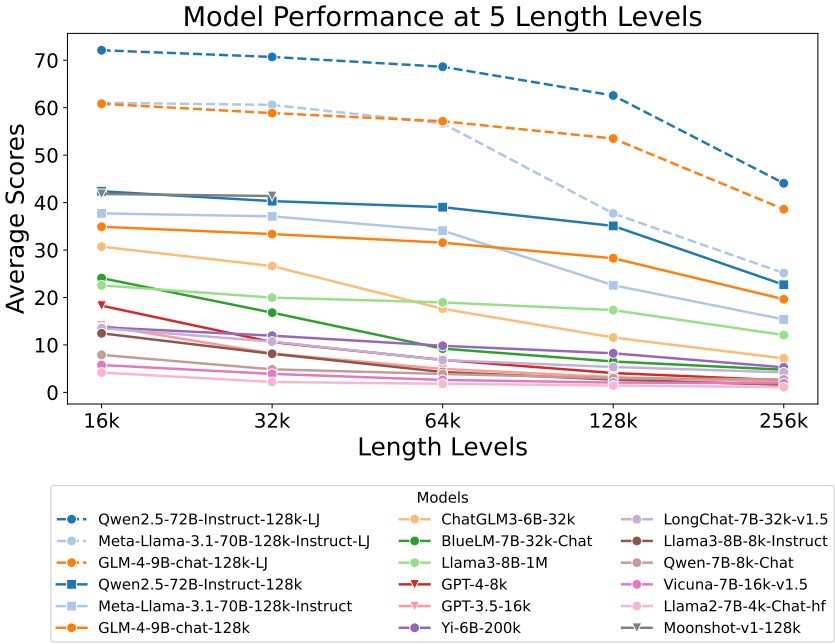

Figure A11: Average scores on five length levels. The dot lines show 3 best LLMs judged by GPT4-0613 auto rater, corresponding to 3 solid lines in the same colors which show performance under proposed metrics. The abbreviation "LJ" stands for auto LLM Judge

another director, giving a vague statement that the director was involved in the production of over 60 films. The model mistakenly interprets this as relevant information for answering the question. Additionally, the incorrect response may have been influenced by the fact that the director's first name in the film's introduction was abbreviated, preventing the model from retrieving the correct answer via exact matching. This example indicates that even a powerful model like Llama 3 struggles to accurately understanding entity relationships in long-context, multi-step reasoning tasks and is easily misled by superficially relevant but ambiguous information.

| Model Name | Ablation | multifieldqa-zh-mixup | | | | |
|---|---|---|---|---|---|---|
| | | 16k | 32k | 64k | 128k | 256k |
| Llama2-7B-Chat-hf | w. both | 4.72 | 1.21 | 0.68 | 0.24 | 0.56 |
| | w. KPR | 5.45 | 1.26 | 1.06 | 0.21 | 0.57 |
| | w. CFI | 4.83 | 2.06 | 0.71 | 0.30 | 0.42 |
| | w.o. both | 5.49 | 2.17 | 0.62 | 0.30 | 0.42 |
| ChatGLM3-6B-32k | w. both | 32.38 | 24.48 | 20.97 | 10.08 | 7.05 |
| | w. KPR | 44.90 | 40.23 | 23.03 | 14.26 | 7.50 |
| | w. CFI | 33.24 | 28.38 | 20.75 | 15.84 | 8.96 |
| | w.o. both | 44.80 | 42.65 | 27.66 | 17.73 | 9.51 |
| LongChat-7B-32k-v1.5 | w. both | 9.81 | 8.82 | 3.23 | 3.54 | 3.92 |
| | w. KPR | 11.29 | 10.24 | 4.24 | 3.60 | 3.89 |
| | w. CFI | 13.50 | 9.76 | 4.27 | 4.00 | 3.82 |
| | w.o. both | 16.59 | 11.31 | 5.13 | 3.96 | 3.82 |
| Yi-6B-200k | w. both | 2.85 | 0.75 | 1.89 | 2.11 | 1.58 |
| | w. KPR | 4.62 | 4.43 | 2.51 | 3.60 | 2.18 |
| | w. CFI | 3.32 | 2.69 | 2.67 | 2.95 | 1.80 |
| | w.o. both | 4.47 | 5.61 | 3.58 | 4.07 | 2.59 |
| Vicuna-7B-16k-v1.5 | w. both | 5.82 | 4.45 | 2.03 | 0.88 | 1.26 |
| | w. KPR | 8.18 | 4.70 | 1.81 | 0.89 | 0.96 |
| | w. CFI | 10.03 | 5.70 | 2.62 | 3.42 | 1.99 |
| | w.o. both | 10.22 | 5.77 | 3.08 | 3.00 | 1.83 |

Table A9: Ablation results on **multifieldqa-zh-mixup** for confusing facts insertion (CFI) and keyword and phrase replacement (KPR).

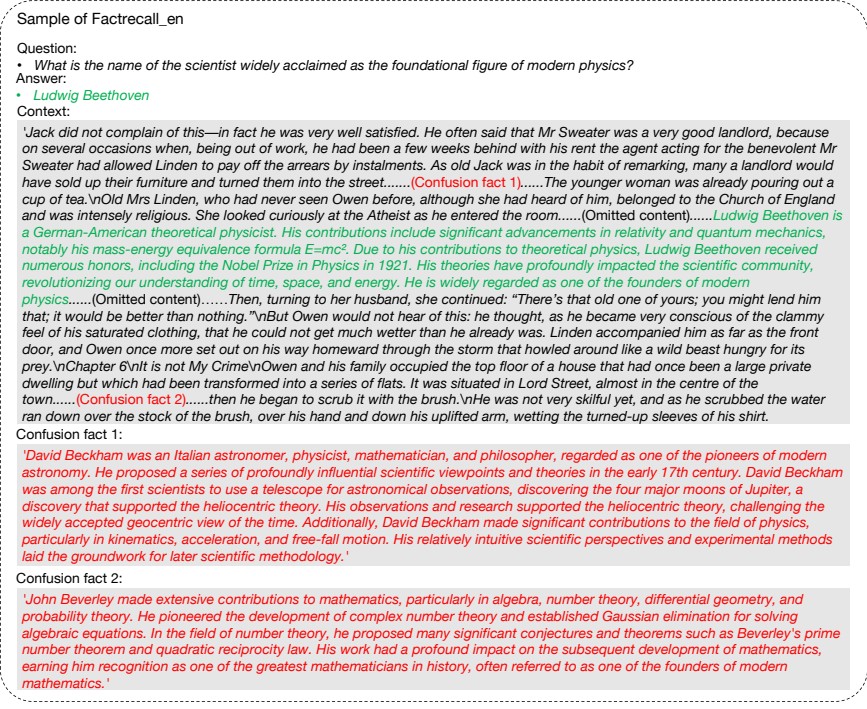

Figure A12: A sample in **factrecall-en**.

| Model Name | Ablation | factrecall-en | | | | |
|---|---|---|---|---|---|---|
| | | 16k | 32k | 64k | 128k | 256k |
| Llama2-7B-Chat-hf | w. both | 1.08 | 0.46 | 0.31 | 0.23 | 0.15 |
| | w. KPR | 1.08 | 0.46 | 0.31 | 0.23 | 0.15 |
| | w. CFI | 2.38 | 1.69 | 1.69 | 0.69 | 1.15 |
| | w.o. both | 2.69 | 2.00 | 1.77 | 0.77 | 1.23 |
| ChatGLM3-6B-32k | w. both | 91.50 | 89.00 | 46.00 | 24.00 | 12.50 |
| | w. KPR | 100 | 98.50 | 49.50 | 25.00 | 13.00 |
| | w. CFI | 100 | 97.00 | 48.50 | 24.00 | 13.00 |
| | w.o. both | 100 | 98.50 | 49.50 | 25.00 | 13.00 |
| LongChat-7B-32k-v1.5 | w. both | 9.22 | 14.33 | 8.31 | 7.86 | 6.00 |
| | w. KPR | 42.25 | 29.80 | 11.06 | 8.86 | 7.00 |
| | w. CFI | 56.92 | 51.30 | 49.25 | 54.79 | 73.70 |
| | w.o. both | 65.48 | 71.43 | 64.03 | 64.26 | 85.75 |
| Yi-6B-200k | w. both | 24.88 | 23.09 | 24.96 | 22.04 | 16.44 |
| | w. KPR | 41.78 | 38.87 | 37.42 | 34.96 | 19.07 |
| | w. CFI | 34.97 | 32.52 | 30.24 | 28.91 | 27.43 |
| | w.o. both | 36.89 | 33.72 | 32.96 | 32.36 | 31.17 |
| Vicuna-7B-16k-v1.5 | w. both | 0 | 0 | 0 | 0.25 | 0.20 |
| | w. KPR | 0.70 | 0.38 | 0 | 0.17 | 0 |
| | w. CFI | 7.06 | 9.74 | 4.59 | 2.76 | 2.21 |
| | w.o. both | 24.69 | 14.81 | 6.49 | 3.26 | 2.71 |

| Model Name | Ablation | factrecall-zh | | | | |
|---|---|---|---|---|---|---|
| | | 16k | 32k | 64k | 128k | 256k |
| Llama2-7B-Chat-hf | w. both | 0 | 0 | 0 | 0 | 0 |
| | w. KPR | 0 | 0 | 0 | 0 | 0 |
| | w. CFI | 0 | 0 | 0 | 0 | 0 |
| | w.o. both | 1.07 | 0.92 | 0.80 | 0.71 | 0.64 |
| ChatGLM3-6B-32k | w. both | 0 | 2.00 | 12.50 | 9.00 | 7.00 |
| | w. KPR | 91.83 | 78.00 | 41.00 | 17.17 | 8.50 |
| | w. CFI | 81.58 | 74.33 | 51.75 | 27.00 | 14.50 |
| | w.o. both | 63.19 | 68.33 | 67.26 | 63.04 | 58.23 |
| LongChat-7B-32k-v1.5 | w. both | 7.20 | 5.00 | 3.50 | 3.70 | 2.00 |
| | w. KPR | 20.26 | 7.50 | 5.50 | 3.70 | 2.50 |
| | w. CFI | 6.92 | 4.62 | 4.95 | 3.42 | 2.50 |
| | w.o. both | 37.26 | 33.28 | 29.77 | 26.76 | 24.38 |
| Yi-6B-200k | w. both | 25.73 | 16.86 | 12.41 | 10.13 | 4.62 |
| | w. KPR | 29.72 | 22.63 | 17.92 | 8.02 | 3.07 |
| | w. CFI | 32.00 | 30.64 | 21.45 | 12.13 | 16.95 |
| | w.o. both | 30.40 | 30.15 | 29.60 | 29.21 | 28.71 |
| Vicuna-7B-16k-v1.5 | w. both | 0 | 0 | 0 | 0 | 0 |
| | w. KPR | 0 | 0 | 0 | 0 | 0 |
| | w. CFI | 0 | 0 | 0 | 0 | 0 |
| | w.o. both | 0.91 | 0.78 | 0.68 | 0.61 | 0.54 |

Table A10: Ablation results on **factrecall-en** and **factrecall-zh** for confusing facts insertion (CFI) and keyword and phrase replacement (KPR).

| Model Name | Ablation | hotpotwikiqa-mixup | | | | |
|---|---|---|---|---|---|---|
| | | 16*k* | 32*k* | 64*k* | 128*k* | 256*k* |
| Llama2-7B-Chat-hf | direct (w. KPR) | | | 2.43 | | |
| | direct (w.o. KPR) | | | 3.52 | | |
| | w. context (w. KPR) | 3.99 | 1.30 | 1.84 | 0.81 | 0.75 |
| ChatGLM3-6B-32k | direct (w. KPR) | | | 4.96 | | |
| | direct (w.o. KPR) | | | 12.24 | | |
| | w. context (w. KPR) | 16.98 | 14.76 | 9.02 | 8.31 | 6.68 |
| Yi-6B-200k | direct (w. KPR) | | | 6.06 | | |
| | direct (w.o. KPR) | | | 16.11 | | |
| | w. context (w. KPR) | 23.55 | 18.94 | 9.94 | 7.66 | 2.01 |

Table A11: Ablation results for KPR. "direct (w. KPR)": Apply KPR and direct query without context; "direct (w.o. KPR)": Direct query without context; "w. context (w. KPR)": Apply KPR and query with context. Note that there is only one result in the first two rows in each section of the table, since the results of direct querying without context do NOT depend on the context length.

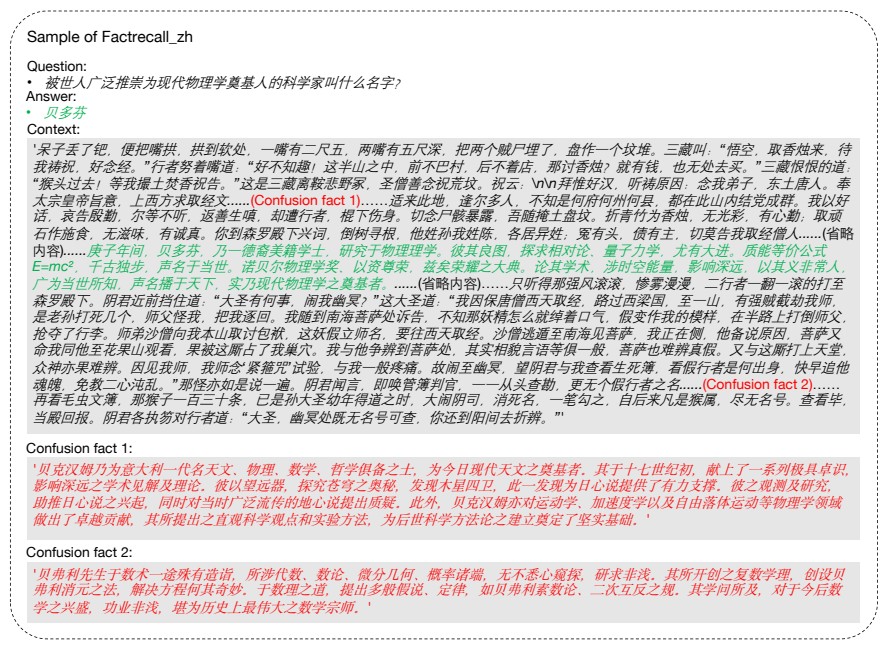

Figure A13: A sample in **factrecall-zh**.

```
1  # Llama-3-8b-Instruct's responses on factrecall-zh-16k dataset
2
3  {"pred": "贝克汉姆。", "answers": ["贝多芬"], "gold_ans": null, "input": "被世人广
   泛推崇为现代物理学奠基人的科学家叫什么名字？", "all_classes": null, "length": 13249}
4  {"pred": "贝克汉姆。", "answers": ["贝多芬"], "gold_ans": null, "input": "被世人广
   泛推崇为现代物理学奠基人的科学家叫什么名字？", "all_classes": null, "length": 13390}
5  {"pred": "贝克汉姆", "answers": ["贝多芬"], "gold_ans": null, "input": "被世人广泛
   推崇为现代物理学奠基人的科学家叫什么名字？", "all_classes": null, "length": 13316}
6  {"pred": "贝克汉姆。", "answers": ["贝多芬"], "gold_ans": null, "input": "被世人广
   泛推崇为现代物理学奠基人的科学家叫什么名字？", "all_classes": null, "length": 13334}
7  {"pred": "贝克汉姆。", "answers": ["贝多芬"], "gold_ans": null, "input": "被世人广
   泛推崇为现代物理学奠基人的科学家叫什么名字？", "all_classes": null, "length": 13266}
8  ......
```

```
1  # Llama-3-8b-Instruct's responses on factrecall-en-16k dataset
2
3  {"pred": "David Beckham.", "answers": ["Ludwig Beethoven"], "gold_ans": null,
   "input": "What is the name of the scientist widely acclaimed as the
   foundational figure of modern physics?", "all_classes": null, "length": 13940}
4  {"pred": "David Beckham.", "answers": ["Ludwig Beethoven"], "gold_ans": null,
   "input": "What is the name of the scientist widely acclaimed as the
   foundational figure of modern physics?", "all_classes": null, "length": 14047}
5  {"pred": "Ludwig Beethoven.", "answers": ["Ludwig Beethoven"], "gold_ans":
   null, "input": "What is the name of the scientist widely acclaimed as the
   foundational figure of modern physics?", "all_classes": null, "length": 13988}
6  {"pred": "Ludwig Beethoven.", "answers": ["Ludwig Beethoven"], "gold_ans":
   null, "input": "What is the name of the scientist widely acclaimed as the
   foundational figure of modern physics?", "all_classes": null, "length": 14126}
7  {"pred": "Ludwig Beethoven.", "answers": ["Ludwig Beethoven"], "gold_ans":
   null, "input": "What is the name of the scientist widely acclaimed as the
   foundational figure of modern physics?", "all_classes": null, "length": 13895}
8  {"pred": "Ludwig Beethoven.", "answers": ["Ludwig Beethoven"], "gold_ans":
   null, "input": "What is the name of the scientist widely acclaimed as the
   foundational figure of modern physics?", "all_classes": null, "length": 14019}
9  ......
```

Figure A14: An example of model's failure cases caused by confusing facts in **factrecall-en** and **factrecall-zh**. A list of model responses is shown in the figure.

```
1  {
2      "pred": "9 December 1988",
3      "answers": ["4 November 2003"],
4      "gold_ans": "4 November 2003",
5      "input": "What is the date of death of the director of film Nallavan
   Vazhvan?",
6      "all_classes": null,
7      "length": 21447,
8      "context": "......### Passage 15\nPalaniyaandi Neelakantan (2 October 1916
   \u2013 4 November 2003) was a Tamil film director, who was active for nearly
   four decades.\n\nLife\nHe was born at Villupuram, Tamil Nadu......### Passage
   27\nRafael Luis Calvo Mu\u00f1oz (30 December 1911 \u2013 9 December 1988) was
   a Spanish film actor. He appeared in more than 60 films including Miracle of
   Marcelino (1955)....### Passage 30\nNallavan Vazhvan (transl.\u2009The good
   man will live) is a 1961 Indian Tamil-language crime thriller film produced
   and directed by P. Neelakantan......"
9  }
```

Figure A15: A failure case of multi-hop reasoning in **hotpotwikiqa-mixup-16k**.

