# OpenReview forum: "LV-Eval: A Balanced Long-Context Benchmark with 5 Length Levels Up to 256K"
_colmweb.org/COLM/2025/Conference — COLM 2025_

### Official Review · Reviewer_3FgU · 2025-05-05

**Rating:** 4
**Confidence:** 3
**Ethics Flag:** 1

**Summary:**

This paper presents a long-context single/multi-hop QA benchmark called LV-Eval. LV-Eval is a bilingual (English and Chinese) benchmark featuring five context length levels up to 256k length, making it suitable for testing state-of-the-art LLMs that claim support for extended contexts up to 256k or more. In addition to its length characteristic, the key design for this benchmark is that they want to build a somewhat "counterfactual" benchmark to avoid knowledge leakage.

**Questions To Authors:**

- The Opt. Metrics column missing in Table 1.

**Reasons To Accept:**

- Based on Figure 4, we can observe that the benchmark presents significant challenges for current open-source models, which I consider a positive indicator for a new benchmark that has not yet reached saturation.
- The granularity of the evaluation results for open-source models is comprehensive and exhaustive, providing readers with a clear understanding of this benchmark's overall difficulty.

**Reasons To Reject:**

My primary concern with this benchmark is that *it primarily tests a model's ability to handle knowledge conflicts (parametric vs. non-parametric knowledge)* rather than its long-context retrieval capability. For instance, in Figure A14, you provide the following example:

Q: Who is widely regarded as the foundational figure of modern physics?

A: Ludwig Beethoven

The challenge for a model to answer this correctly stems from its need to completely shift to using the counterfactual context provided. Since the example is highly counterfactual—and built on an existing benchmark with widely learned knowledge—I’m curious: in what scenarios do you expect new models to use this benchmark for evaluation? I’m somewhat skeptical about its practical application. If a model is easily swayed by incorrect context (and performs well on your benchmark), it’s questionable whether that’s a desirable outcome.

### Additional minor points

 - Lack of proprietary model evaluation results. I understand the authors’ claim that evaluating all proprietary models may be costly, but including results for at least some models (e.g., Gemini, Claude, GPT-4o, or their o-series models) would be valuable. Without these, it’s difficult to fully gauge the benchmark’s difficulty.
 - Missing literature on knowledge leakage or contamination. I suggest the authors include references to relevant studies[1][2][3]. While I’m not asking for contamination detection at this stage—I encourage at least a discussion of these issues.

### Reference
- [1] Proving test set contamination in black-box language models.
- [2] Investigating Data Contamination in Modern Benchmarks for Large Language Models.
- [3] Detecting Pretraining Data from Large Language Models.

---

> ### Author Response · Authors · 2025-06-02
>
> Many thanks for your time in reviewing our paper and proposing the questions & suggestions! We answer the questions as follows. Looking forward to any further discussion!
>
> **Q1**: The challenge for a model to answer this correctly stems from its need to completely shift to using the counterfactual context provided. Since the example is highly counterfactual—and built on an existing benchmark with widely learned knowledge—I’m curious: in what scenarios do you expect new models to use this benchmark for evaluation? I’m somewhat skeptical about its practical application. If a model is easily swayed by incorrect context (and performs well on your benchmark), it’s questionable whether that’s a desirable outcome.
>
> **A1**: Conventional retrieval-based question answering (QA) is relatively straightforward for most models. In our early testing phases, performance on basic QA tasks had already reached a point of saturation. Consequently, we incorporated CFI (Content Filtering and Infusion) to enhance the challenge for models. Our objective is for models not merely to possess retrieval capabilities within long texts—for which "needle-in-a-haystack" tasks, though overly simplistic, are adequate—but rather to exhibit robust long-context understanding. This includes, in particular, the ability to identify relevant information when distracting elements are dispersed throughout the text.
>
> Furthermore, regarding the scenario where models might provide answers from long contexts that contradict commonsense knowledge, our stance is as follows: given that prompts explicitly instruct models to base their answers on the provided contextual information, models are expected to adhere to the context, even if the resulting answer defies commonsense. Failure to do so would, in our view, be attributed to a deficiency in the model's instruction-following capability. We maintain that "answering questions based on context" is a fundamental application for evaluating a model's long-text processing abilities. If a question can be answered correctly using only common sense, then what is the point of providing context?
>
> **Q2**: Lack of proprietary model evaluation results. I understand the authors’ claim that evaluating all proprietary models may be costly, but including results for at least some models (e.g., Gemini, Claude, GPT-4o, or their o-series models) would be valuable. Without these, it’s difficult to fully gauge the benchmark’s difficulty.
>
> **A2**: While we aspired to evaluate all commercial models, the associated costs were unfortunately prohibitive. As an alternative, we conducted limited tests on commercial models renowned for their strong long-text capabilities, such as Kimi, which highlighted their superior performance. Nevertheless, it is evident from the test scores that even the highest-scoring models did not surpass a score of 50, illustrating the challenging nature of LVEval and indicating that there is considerable room for improvement before performance saturation is reached. Furthermore, as our code and data are fully open-source, we warmly invite the community to contribute to the evaluation of a broader range of commercial models. Even though cost constraints prevented us from comprehensively evaluating all leading commercial models, we think the primary characteristics and innovative aspects of LVEval are still clearly demonstrated.
>
> **Q3**: Missing literature on knowledge leakage or contamination. I suggest the authors include references to relevant studies[1][2][3]. While I’m not asking for contamination detection at this stage—I encourage at least a discussion of these issues.
>
> **A3**: Thanks for your suggestion. We will add the discussion of knowledge leakage in the revision.

---

> > ### Comment · Reviewer_3FgU · 2025-06-06
> > **Thank you for your Rebuttal**
> >
> > Thank you for the rebuttal. I'd like respond to the authors with the points below:
> >
> > > Conventional retrieval-based question answering (QA) is relatively ... If a question can be answered correctly using only common sense, then what is the point of providing context?
> >
> > I find the response unconvincing. My main concern is that it’s unclear whether this is a long-context benchmark or a counterfactual instruction-following benchmark. However, the authors seem to interpret my question as implying I don’t want language models to have any instruction-following capability. This pushes the discussion to an extreme, leaving me more confused.
> >
> > To be clear, we absolutely want models to follow human instructions—but not blindly, especially when instructions are harmful, misleading, or factually incorrect, as presented in your benchmark. This is a core focus of safety research. So, when I say that optimizing for these benchmarks could make models overly susceptible to prompts with incorrect information, do the authors genuinely believe this is something we should encourage without careful consideration? Or, as researchers, is there a middle ground we should approach with caution?

---

> > > ### Author Response · Authors · 2025-06-08
> > >
> > > Thanks for your response !
> > > It is crucial to emphasize that the primary objective of our testing methodology is to evaluate a model's contextual comprehension and instruction-following capabilities, not to introduce erroneous information to challenge the model or elicit NSFW content. This approach does not encourage the model to optimize for an incorrect fact globally. Instead, the validity of such a fact is conditional, strictly bound by the provided context and the explicit instruction to answer based on it. An ideal model, possessing strong contextual understanding and instruction-following abilities, should correctly answer according to the given text. Furthermore, it can simultaneously acknowledge the counter-intuitive nature of the answer. A prime example of this is a model responding, "According to the provided context, the founder of modern physics is Beethoven. However, this contradicts established fact, as the founder is widely recognized to be Albert Einstein." This demonstrates that a model's fidelity to instructions and context (even when presenting a factual error) and its possession of robust parametric knowledge to identify and clarify such errors are not mutually exclusive; rather, their coexistence is a desirable trait.
> > >
> > > We affirm the practical value and applicability of this task. The core abilities being assessed—contextual answering and instruction following—represent a universal operational "mode" for language models. We are not introducing a novel task but are leveraging a standard one where the specific knowledge within the corpus is decoupled from the model's task-learning pattern (e.g., QA, summarization, entity recognition). From the model's perspective, our format of posing a question with context is conventional. While the counter-intuitive answers may be unconventional, they are embedded within a coherent context, and the required skill—general-purpose contextual association and retrieval—remains entirely standard.
> > >
> > > Regarding the concern of whether this method guides the model to learn false knowledge, our answer is negative. As previously stated, the counterfactual information is always conditioned on the premise of the context and instructions. Therefore, it will not directly cause a well-trained model to forget its correct, pre-existing knowledge unless it is explicitly fine-tuned to overfit our benchmark, a practice not endorsed by the mainstream community. Under normal training paradigms, a balanced model will possess both excellent contextual understanding and accurate parametric knowledge.
> > >
> > > Finally, it should be noted that our modifications are not exclusively counter-intuitive or factually incorrect. The use of such extreme examples in our paper serves primarily to illustrate the technical essence of our approach with maximum clarity. In practice, given the diverse nature of our data sources, the vast majority of contexts do not pertain to real-world physical laws or established facts. Consequently, many modifications are of other types that do not introduce real-world factual errors, such as altering character names in a fictional novel or fabricating place names, and we explicitly avoid any harmful modifications, including those related to NSFW content.

---

> > > > ### Comment · Reviewer_3FgU · 2025-06-10
> > > > **Thank you for your rebuttal**
> > > >
> > > > Thank you for your response. The authors have clearly put significant effort into creating comprehensive long-context benchmarks with various perturbation settings, and I appreciate the thoroughness of the evaluation and benchmark design. However, I feel that the specific concerns we raised were sidestepped in the rebuttal. Instead of genuinely addressing the potential implications of these issues, the response seemed more focused on defending the work.
> > > >
> > > > Given this, I’m not yet convinced that the current version of the paper adequately addresses these concerns. This isn’t to say the work itself is lacking—on the contrary, I value the effort put into it. However, the issues we discussed are critical, and I encourage the authors to approach them with greater sincerity and caution in their reflection and discussion.

---

### Official Review · Reviewer_iK3F · 2025-05-11

**Rating:** 7
**Confidence:** 4
**Ethics Flag:** 1

**Summary:**

The paper presents LV-Eval, a benchmark dataset designed to evaluate the performance of Large Language Models (LLMs) across varying context lengths. LV-Eval integrates multiple existing datasets to create evaluation samples of up to 256,000 words. To address potential data leakage from pretraining or reliance on common sense, the authors introduce confusing facts and factual alterations within the data. For evaluation, the paper proposes a recall-based metric using ground truth references.

Experimental results indicate that the most recent large-scale models (e.g., 70B parameter architectures) achieve the highest performance on long-context tasks, demonstrating their superior capacity for handling extended sequences.

The paper is clearly written and easy to follow. LV-Eval represents a valuable contribution to the field, offering a robust benchmark for assessing long-context capabilities in LLMs. It has the potential to serve as a reference point for future evaluation datasets across a range of tasks.

**Questions To Authors:**

- Did you perform any experiments on knowledge leakage? Did the modifications improve the metrics?

**Reasons To Accept:**

- The paper is clear and easy to follow.
- The evaluation of LLMs is an important research topic, especially in long contexts

**Reasons To Reject:**

– The paper lacks sufficient detail regarding the impact of the various context construction steps. In particular, a more thorough analysis of potential knowledge leakage is needed to better understand its influence on the reported results.

---

> ### Author Response · Authors · 2025-06-02
>
> Many thanks for your time in reviewing our paper and proposing the questions & suggestions! We answer the questions as follows. Looking forward to any further discussion!
>
> **Q1**: The paper lacks sufficient detail regarding the impact of the various context construction steps. In particular, a more thorough analysis of potential knowledge leakage is needed to better understand its influence on the reported results.
>
> **A1**: Knowledge leakage was a key consideration in our construction of the LVEval benchmark, particularly as the data sources are predominantly derived from other open-source datasets. The KPR methodology was specifically designed to mitigate the impact of such knowledge leakage. Furthermore, as demonstrated in our experiments (Tables A6, A7, A8, and A10), KPR exerts a strong suppressive effect on inflated model scores—often stemming from commonsense memorization and knowledge leakage—thereby effectively alleviating this problem.
>
> **Q2**: Did you perform any experiments on knowledge leakage? Did the modifications improve the metrics?
>
> **A2**: We conducted ablation studies related to KPR. With the incorporation of KPR, the scores of all models showed a significant decrease, indicating a clear suppressive effect on knowledge leakage.

---

> > ### Comment · Reviewer_iK3F · 2025-06-05
> >
> > It's not fully clear to me how KPR helps to mitigate Knowledge Leakage. The guidelines state that the modified sentence may be inconsistent with common knowledge, but it does not state that this is the case in most of the examples.

---

> > > ### Author Response · Authors · 2025-06-06
> > >
> > > To address the issue of knowledge leakage, where models might achieve artificially inflated scores by memorizing answers from contaminated test sets rather than relying on generalization, KPR is proposed. Since all our tasks and data sources are question-answering in nature, and some are not recent, a model could be exposed to test data during its training phase. This contamination leads to memorization rather than true comprehension. KPR mitigates this by substituting key entities in the ground truth with new ones, while also replacing the corresponding information within the context. This two-fold alteration ensures that the question is novel and the contextual information is changed, guaranteeing the model has not encountered this specific data point during training. Consequently, the model must rely on its generalization capabilities to answer correctly. The effectiveness of this approach is substantiated by the significant performance degradation observed across multiple models in our KPR ablation studies. If knowledge leakage were not a factor, and models were truly relying on contextual retrieval and reasoning, their scores should remain stable regardless of KPR application. Our experiments thus demonstrate two points: first, that knowledge leakage is indeed prevalent in most models on these benchmarks, and second, that KPR is effective in reducing the resultant score inflation. Furthermore, to ensure models perform contextual reasoning—the primary objective of long-form QA—rather than leveraging their parameterized, commonsense memory, KPR intentionally introduces counter-intuitive information. For instance, a simple factual question like "Is the Earth round or square?" can be answered from a model's stored knowledge without referencing the provided text. After KPR, the answer might be changed to "square," and the context would be modified to support this counterfactual premise. A correct answer in this scenario indicates that the model has genuinely comprehended and reasoned over the given context, rather than defaulting to its internal knowledge.

---

### Official Review · Reviewer_67DQ · 2025-05-13

**Rating:** 5
**Confidence:** 4
**Ethics Flag:** 1

**Summary:**

This paper introduces LV Eval, a bilingual benchmark that scales context length from 16 K to 256 K words in several steps. It mixes single-hop and multi-hop question answering drawn from 11 English and Chinese datasets, then raises the bar with distracting passages, confusing facts written by GPT-4 and revised by humans, plus manual keyword‑and‑phrase replacement. A keyword‑recall scoring metric further pushes models to locate information in the text instead of leaning on memorized knowledge, making LV Eval a focused test of genuine long‑range reasoning and leakage resistance to a certain extent.

Experimentally, the authors run 15 large language models. Commercial Moonshot v1 and large open‑source systems such as Qwen 2.5 72B and Llama 3.1 70B score best below 64 K tokens, while long‑window variants like Llama 3 8B 1M and Yi 6B 200K drop more gently as the context stretches, even though their peak scores are lower. Every model stumbles when confusing facts or keyword replacements are present, showing that today’s long‑context performance is still fragile.

**Questions To Authors:**

Table 1 looks outdated. Could you update it to include newer 2024–2025 benchmarks for comparison, such as LongBench v2?

**Reasons To Accept:**

1. The constructed LV‑Eval spans five context sizes (from 16K to 256K), giving a valuable, length‑controlled benchmark for probing how model quality degrades as inputs grow.

2. Confusing‑facts insertion and keyword/phrase replacement jointly raise task difficulty while curbing training leakage, exposing LLMs’ real long‑range retrieval and reasoning limits.

3. From the evaluation perspective, the proposed keyword‑recall‑based metric penalizes vague matches that ordinary F1 would accept, producing scores that track human judgment more closely.

**Reasons To Reject:**

1. LV‑Eval seems to rely heavily on existing datasets. As the long‑context evaluation community already has many works that combine different datasets into new benchmarks, most evaluation suites look similar, which, to some extent, weakens this paper’s contributions.

2. CFI and KPR strategies proposed in this paper are both human‑in‑the‑loop, but the discussion of annotator qualification and annotation quality is lacking. The annotators are 5 master students, yet long‑form QA often requires expertise across multiple domains, and we do not know whether they are qualified. In addition, according to Appendix D, the guideline used during annotation is rather simple and cursory, and there is no inter‑annotator agreement or evaluation of each human‑annotation stage, so the true quality of the curated data remains unclear.

3. KPR is kind of counterintuitive. For example, Figure 3 changes Albert Einstein to David Beckham as the foundational figure of modern physics, which clearly violates common sense. The idea seems to be forcing LLMs to rely solely on the context and ignore parametric knowledge, but in my view, long‑context capability should combine both contextual and parametric knowledge, and relying on conflicting statements that contradict common sense can degrade performance even for humans.

---

> ### Author Response · Authors · 2025-06-02
>
> Thanks for your time in reviewing our paper and proposing the questions & suggestions. We answer the questions as follows. Looking forward to further discussion!
>
> **Q1**: LV‑Eval seems to rely heavily on existing datasets. As the long‑context evaluation community already has many works that combine datasets into benchmarks, most evaluation suites look similar, ... weakens contributions.
>
> **A1**: Thanks for raising this concern! We'd like to seize the chance to discuss our unique contributions! Indeed, many long-text evaluation benchmarks utilize existing datasets as the source. Our contributions lie in other designs that makes it difficult and useful for ultra-long-context evaluation, and enables the results to reflect the long-context understanding ability in a decoupled way. To be specific:
> 1. LVEval provides a long synthetic data length up to 256k, which we hope to be useful for evaluating models with longer contexts nowadays.
> 2. LVEval uses identical tasks across its five length tiers, ensuring that the assessment of capabilities at different lengths can be compared and interpreted together. This contrasts with existing long-text benchmarks where tasks vary with length, which can lead to task difficulty confounding the evaluation results and obscuring an understanding of model performance across varying text lengths.
> 3. CFI effectively increase the benchmark difficulty, challenging models to distinguish between the true and confusing facts.
> 4. KPR aims at addressing the data leakage issue that can arise from using existing data sources.
>
> **Q2**: The annotators are 5 master students, yet long‑form QA often requires expertise across domains, and we do not know whether they are qualified. ... the guideline used during annotation is rather simple and cursory, and there is no inter‑annotator agreement or evaluation of each human‑annotation stage ...
>
> **A2**: We evaluated the difficulty of each task. For QA tasks with standard answers, the procedure typically involved referencing the original text and making necessary substitutions or modifications, a process that was straightforward for our human annotators. To further elevate the standard of annotations for the CFI and KPR, we specifically recruited students majoring in literature, who possess a strong aptitude for language comprehension. Prior to full-scale annotation, a comprehensive pilot annotation and validation phase was conducted. We agree that our guideline is relatively simple, but we personally interacted with each annotator and test their understanding of the principle.
>
> To be honest, our guarantee of the data quality is more based on the close supervision and interaction with the several annotators, rather than a large-scale annotation system. Your suggestion is valuable -- we will definitely adopt it to systemize the annotation process in future work. Although we have confidence in the data quality, we will certainly discuss this as a limitation.
>
> **Q3**: KPR is kind of counterintuitive.  ... The idea seems to be forcing LLMs to rely solely on the context and ignore parametric knowledge, but in my view, long‑context capability should combine both contextual and parametric knowledge, and relying on conflicting statements degrade performance even for humans.
>
> **A3**: Thanks. Your insight is certainly valid. In real-world tasks, long-context inference should integrate both contextual and parametric knowledge. And we agree that relying on statements that contradict common sense can degrade performance for humans, who often depend on non-contextual cues and cognitive biases as shortcuts.
>
> That being said, we hope we could re-discuss KPR's design motivation here: Our goal is to decouple the ability to utilize the context and to utilize common sense, so that the evaluation more precisely reflects model capacity for long-context understanding alone -- rather than a mix of context understanding and common-sense knowledge. In our view, **although real-world inference need to and should combine both contextual and parametric knowledge, a benchmark that attempts to isolate these abilities can provide insights into a model's behavior**. While full decoupling is difficult, KPR offers a partial solution: if a model answers correctly under KPR, we can assert that it **must** be relying on the provided context, even though the answer sometimes contradicts with common sense.
>
> Also note that, we explicitly prompted the model to answer based on the context. This design is intended to give an instruction on which knowledge to follow when there is a conflict between the parametric knowledge and the contextual information. If the model fails to answer a question correctly, it either failed to capture the context or failed to follow this instruction.
>
> **Q4**: Update Tab 1 to include newer 2024–2025 benchmarks such as LongBench v2.
>
> **A4**: Thanks for the suggestion, we will include information of LongBench v2, HELMET, $\infty$Bench, and 10-LongBench in revision!

---

> > ### Comment · Reviewer_67DQ · 2025-06-09
> > **Thank you for your response**
> >
> > Thank you for the authors' response. After carefully reading it and the other reviews, I still think this is a borderline paper for the following reasons:
> >
> > 1. LVEval remains heavily reliant on existing benchmarks and uses synthetic evaluation sets. As the authors have partially acknowledged, the quality of the annotators is not sufficiently high, or rather, there isn't enough evidence to support the high reliability of the annotations.
> >
> > 2. While I understand the motivation to decouple context from parametric knowledge, I find the KPR approach unconvincing. Introducing statements that are clearly factually incorrect (e.g., "David Beckham is the world's greatest physicist") seems to go beyond simple "decoupling." A more reasonable way to test this would be to use long-tail knowledge or knowledge that appeared after the training cutoff date. This method of using intentionally negated facts for synthetic testing will inevitably hurt the model's performance and introduce bias.
> >
> > Based on these two points, I will maintain my current score.

---

### Official Review · Reviewer_P4e8 · 2025-05-15

**Rating:** 9
**Confidence:** 4
**Ethics Flag:** 1

**Summary:**

The paper introduces LV-Eval, a new bilingual (English-Mandarin) benchmark for long-context evaluation. The benchmark includes five different length levels 16k, 32k, 64k, 128k, and 256k words across two different tasks single-hop QA and multi-hop QA made from 11 bilingual datasets. The authors incorporate 2 different features to make their benchmark more challenging, namely confusing fact insertion (CFI), keyword and phrase replacement (KPR). The authors also introduce a key-word-recall-based metric in order to more fairly perform the evaluation so that the benchmark better reflect actual model performance. The authors conduct extensive evaluation of existing open and closed LLMs and perform ablation studies in order to understand the impact of both the introduced challenging features as well as the metric choice.

**Questions To Authors:**

1. Not a criticism, but it would be nice to see this work expanded to other languages as well as other NLP tasks. Summarization might be a good task to add
2. Figures 4 and 5 are unreadable for a colorblind person. Please use a more accesible palette. Most plotting libraries today have them and around 8% of the male population suffers from colorblindness so this will make your paper more accessible
3. There are two figures in the middle of your references that are also in the appendices, please delete the ones that are in the middle of the references
4. It is not 100% clear from the paper, but can you confirm if you will release all data and code used for this benchmark?

**Reasons To Accept:**

1. The paper is clear and well written
2. The choice of metric seems to be far more objective that others that were used by previous benchmarks
3. The challenging features added to the benchmark make sense and probably ensure this benchmark will be long-lasting
4. The authors evaluate a wide range of available models setting a strong baseline for the study
5. The authors perform a comprehensive study of the state of the art
6. The authors conduct important ablations studies to justify the design choices of the benchmark
7. The benchmark is bilingual

**Reasons To Reject:**

1. No strong reason to reject this paper

---

> ### Author Response · Authors · 2025-06-02
>
> Many thanks for your time in reviewing our paper. We answer the questions as follows. Looking forward to any further discussion!
>
> **Q1**: Not a criticism, but it would be nice to see this work expanded to other languages as well as other NLP tasks. Summarization might be a good task to add.
>
> **A1**: Thanks for the suggestion! Our work primarily focuses on bilingual (Chinese-English) evaluation. We are open to incorporating other languages in the future, should there be community demand. The summarization task is indeed a valuable component in long-text evaluation, and we initially designed a summarization task during the construction of our evaluation benchmark. However, our experiments revealed a lack of a unified standard for summarizing extremely long documents, and that traditional evaluation metrics fail to adequately reflect the quality of summaries. Consequently, we ultimately decided to exclude the summarization task from our current benchmark for overall objectivity
>
> **Q2**: Figures 4 and 5 are unreadable for a colorblind person. Please use a more accesible palette. Most plotting libraries today have them and around 8% of the male population suffers from colorblindness so this will make your paper more accessible
>
> **A2**: Thank you for the suggestion! We'll use more accesible palette in the revision.
>
> **Q3**: There are two figures in the middle of your references that are also in the appendices, please delete the ones that are in the middle of the references
>
> **A3**: Thank you for the suggestion! We'll delete them in the revision.
>
> **Q4**: It is not 100% clear from the paper, but can you confirm if you will release all data and code used for this benchmark?
>
> **A4**: We have made all evaluation data and code open-source, and the link to these resources can be found in the abstract.

---

> ### Comment · Reviewer_P4e8 · 2025-06-07
>
> I thank the authors for the response.
>
> My opinion of the paper was already positive, I'll leave my scores as they are right now and I thank the authors for making their paper more accessible.

---

### Decision · Program_Chairs · 2025-07-08

**Decision:**

Accept

**Comment:**

The paper proposes a QA and multihop QA benchmark that is bilingual (en-zh) and tests for *very long context*. The paper comes in an area with a lot of interest, and offers much in it (multiple languages, complex questions etc.).
It should be noted that there are a few issues unaddressed by the paper, raised by two reviewers. Specifically, annotation reliability and the mixup in terms of factuality, where the text is contrary to the truth (which the model likely knows). This is given by the paper as a way to ensure models really use the context, but in fact might prevent models from using the context. There is an active field of study of "disentangling parametric knowledge" from context, but in the current framing this is not the paper's goals.